# Alix is required for activity-dependent bulk endocytosis at brain synapses

**Marine H. Laporte** [1,2�]*, **Kwang Il Chi** [1☯], **Laura C. Caudal** [3], **Na Zhao** [3], **Yvonne Schwarz** [4], **Marta Rolland** [1], **José Martinez-Hernandez** [1,5], **Magalie Martineau** [6], **Christine Chatellard** [1,7], **Eric Denarier** [1], **Vincent Mercier** [1,8], **Florent Lemaître** [1,9], **Béatrice Blot** [1], **Eve Moutaux** [1], **Maxime Cazorla** [1,10], **David Perrais** [6], **Fabien Lanté** [1], **Dieter Bruns** [4], **Sandrine Fraboulet** [1,11], **Fiona J. Hemming** [1], **Frank Kirchhoff** [3]*, **Rémy Sadoul** [1,7]*

1 Grenoble Institute Neurosciences, University Grenoble Alpes—INSERM U1216—CEA, Grenoble, France, 2 Department of Cell Biology, University of Geneva, Geneva, Switzerland, 3 Molecular Physiology, Center for Integrative Physiology and Molecular Medicine (CIPMM), University of Saarland, Homburg, Germany, 4 Molecular Neurophysiology, Center for Integrative Physiology and Molecular Medicine (CIPMM), University of Saarland, Homburg, Germany, 5 Synaptic Structure Laboratory, Instituto de Investigación en Discapacidades Neurológicas (IDINE), Universidad Castilla-La Mancha, Albacete, Spain, 6 Interdisciplinary Institute for Neuroscience, University of Bordeaux-CNRS UMR 5297, Bordeaux, France, 7 Institut de Biologie Structurale, University Grenoble Alpes—CNRS-CEA UMR 5075, Grenoble, France, 8 Department of Biochemistry, University of Geneva, Geneva, Switzerland, 9 Département des Neurosciences, Centre de Recherche du centre hospitalier de Montréal, Canada, 10 Institut de Neuroscience la Timone, CNRS UMR7289, Marseille, France, 11 Institute for Advanced Biosciences, Univ. Grenoble Alpes—INSERM U1209—CNRS UMR 5309, Grenoble, France

☯ These authors contributed equally to this work.
* marine.laporte@unige.ch (MHL); frank.kirchhoff@uks.eu (FK); remy.sadoul@univ-grenoble-alpes.fr (RS)

**Data Availability Statement:** All relevant data are within the paper and its Supporting Information files.

**Funding:** This work was funded by France Alzheimer (AAP CM 2015—388)(R.S.), ANR (ANR-

## Abstract

In chemical synapses undergoing high frequency stimulation, vesicle components can be retrieved from the plasma membrane via a clathrin-independent process called activity-dependent bulk endocytosis (ADBE). Alix (ALG-2-interacting protein X/PDCD6IP) is an adaptor protein binding to ESCRT and endophilin-A proteins which is required for clathrin-independent endocytosis in fibroblasts. Alix is expressed in neurons and concentrates at synapses during epileptic seizures. Here, we used cultured neurons to show that Alix is recruited to presynapses where it interacts with and concentrates endophilin-A during conditions triggering ADBE. Using Alix knockout (ko) neurons, we showed that this recruitment, which requires interaction with the calcium-binding protein ALG-2, is necessary for ADBE. We also found that presynaptic compartments of Alix ko hippocampi display subtle morphological defects compatible with flawed synaptic activity and plasticity detected electrophysiologically. Furthermore, mice lacking Alix in the forebrain undergo less seizures during kainate-induced status epilepticus and reduced propagation of the epileptiform activity. These results thus show that impairment of ADBE due to the lack of neuronal Alix leads to abnormal synaptic recovery during physiological or pathological repeated stimulations.

R11117CC, J.M-H; ANR-17-CE16-0012, M.M and D.P), Ministère de l'Enseignement Supérieur et de la Recherche (K.I.C, M.R. and M.H.L.), Marie Sklodowska-Curie post-doctoral fellowship (H2020-MSCA-IF-2014, M.M.), EC H2020 (MSCA-ITN EU-GliapHD #722053, L.C.C. and F.K.) and FRM (FDT201805005310, E.M). The funders had no role in study design, data collection and analysis, decision to publish, or preparation of the manuscript.

**Competing interests:** The authors have declared that no competing interests exist.

**Abbreviations:** ACSF, artificial cerebrospinal fluid; ADBE, activity-dependent bulk endocytosis; Alix, ALG-2-interacting protein X; CGN, cerebellar granular neuron; CIE, clathrin-independent endocytosis; cko, conditional ko; CME, clathrin-mediated endocytosis; DIV, day in vitro; EEG, electroencephalogram; EPSC, excitatory postsynaptic current; fEPSP, field excitatory postsynaptic potential; FIB-SEM, focused ion beam-scanning electron microscopy; HRP, horseradish peroxidase; IQR, interquartile range; ko, knockout; LTP, long-term potentiation; MEA, multielectrode array; PSD95, postsynaptic density protein 95; ROI, regions of interest; RRP, readily releasable pool; RT, room temperature; SE, status epilepticus; sEPSC, spontaneous excitatory postsynaptic current; SV, synaptic vesicle; syp-pH, synaptophysin-pHluorin; STD, short-term depression; wt, wild type.

## Introduction

Neuronal communication in mammalian brain relies heavily on the activity-dependent release of chemical neurotransmitters from presynaptic boutons. Following fusion of synaptic vesicles (SVs) with the presynaptic membrane, SV lipids and proteins are retrieved by endocytosis. Endocytosis avoids detrimental increase in the plasma membrane surface and allows recycling of the SV components to replenish the SV pool [1]. At moderate levels of stimulation, retrieval of membrane involves clathrin-mediated (CME) and clathrin-independent endocytosis (CIE) in proportions that are still highly debated [2]. Moreover, long-lasting high frequency stimulations also lead to the clathrin-independent internalization of large stretches of presynaptic membranes. This calcium-dependent process, first discovered at the amphibian neuromuscular junction [3], is referred to as activity-dependent bulk endocytosis (ADBE). It is meant to avoid abnormal increase of the synaptic bouton surface and to allow replenishment of SVs during sustained synaptic stimulations [4,5]. Thus, ADBE has been suggested to play key regulatory roles in physiological or pathological events like epilepsy, which are triggered and sustained by high frequency neuronal activity. However, decrypting the physiological role of ADBE has been hindered by the lack of identified molecules that are both specific and essential to this endocytosis mode.

We have recently demonstrated that ALG-2-interacting protein X (Alix) is essential for clathrin-independent bulk endocytosis in fibroblasts [6]. Alix knockout (ko) mice have normally organized but smaller brains [7,8], a phenotype linked with an alteration of CIE in developing neurons [8]. In the adult brain, Alix is ubiquitously expressed but concentrates at hippocampal presynaptic terminals during epileptic seizures [9]. Alix is a cytosolic protein first identified through its calcium-dependent binding to the penta-EF-hand protein ALG-2 (apoptosis-linked gene 2) that helps opening and activating the protein [10–12]. Its activation leads to interaction with membranes as reported in the case of plasma membrane wounds, where ALG-2 binds to inflowing calcium and helps recruiting Alix to the membrane where the latter organizes repair [13]. Alix also interacts with lipids [14] and with several membrane modifying proteins among which endophilin-A proteins (A1, A2, and A3) [11]. These cytoplasmic proteins that contain an N-BAR (Bin/Amphiphysin/Rvs) domain capable of sensing and generating membrane curvature [15] are major actors of CME at synapses [16,17]. They also drive CIE in fibroblasts [18,19] and were shown to control the fast mode of CIE at ribbon synapses [20] as well as in hippocampal neurons [21,22].

The role of Alix in bulk endocytosis in fibroblasts, its capacity to interact with endophilin-A and to be recruited by calcium at membranes, together with its increased concentration at hippocampal synapses during kainate-induced seizures, brought us to test its possible function in ADBE.

Using cultured wild-type (wt) neurons, we now bring evidence that sustained synaptic activity leads to calcium-dependent recruitment of ALG-2 at synapses. ALG-2 in turn interacts with Alix, which binds and concentrates endophilin-A. This protein complex is necessary for ADBE, which is selectively impaired in Alix ko neurons. We also report that synapse morphology and function are both altered in Alix ko brains. This finding correlates well with impairments in synaptic recovery following prolonged stimulation that we detected in hippocampal slices of mice in which Alix was selectively deleted in forebrain neurons. In these mice, the number of seizures during status epilepticus (SE) induced by intracortical kainate injections was reduced as well as the propagation of epileptiform activity to the contralateral side of injection. Thus, our results show that some molecular mechanisms involved in ADBE may also be involved in certain aspects of synaptic plasticity such as short-term synaptic depression and recurrence of epileptic seizures.

## Results

### Alix is recruited to synapses upon synaptic activation

We used dissociated cortical neuron cultures to decipher the role of Alix at synapses. Western blot analysis during *in vitro* differentiation revealed that Alix expression strongly increases during synaptogenesis as indicated by the parallel rise in postsynaptic density protein 95 (PSD95) expression (**Fig 1A**). Moreover, synaptosome-enriched membranes prepared from mature cortical neurons contained Alix (**Fig 1B**). This synaptic pool increased when neuron cultures were incubated 15 min with the GABA$_A$ receptor antagonist bicuculline, together with a weak potassium channel blocker 4-aminopyridine (Bic/4AP) (**Fig 1B**), a treatment known to increase the frequency of action potential bursts and thereby induce sustained intracellular calcium elevation (**S1A and S4E Figs**) [23,24]. Therefore, this observation suggests that Alix tends to concentrate at synapses undergoing prolonged stimulations. To further confirm this, we used live imaging to follow mCherry-Alix relocalization to synapses in mature hippocampal neurons. While the fluorescent signal was homogeneously distributed throughout the entire neuronal cytoplasm in resting conditions (**Fig 1C** and **1D**, t = 0 min), it almost doubled in discrete spots within neuronal processes during synaptic activation (**Fig 1C** and **1D**, t = 2 min 20 s arrowheads). These spots correspond to active presynaptic boutons as they were labeled with synaptophysin-pHluorin (Syp-pH), which becomes fluorescent upon exocytosis at synapses [25] (arrowheads **Fig 1C** and **S1 Movie**). Alix was recruited to 80% of synapses activated by Bic/4AP revealed by a significant Syp-pH increase (**Fig 1E**). Interestingly, Alix recruitment was concomitant with synaptic activation, being detectable a few seconds after addition of Bic/4AP to the culture medium and decreasing soon after (**Fig 1C and 1D**). Alix relocalization was also detected using Alix-YFP, which increased during Bic/4AP treatment at sites confirmed to be at axonal boutons by synapsin-1 immunolabeling (**Fig 1F and 1G**). Significant increase occurred at 37% of the synapsin-positive synapses. Anti-PSD95 immunostaining showed that Alix-YFP positive spots were juxtaposed to, but did not overlap with the postsynaptic density marker (**Fig 1H–1J**). Accordingly, measuring Alix-YFP fluorescence intensities in dendritic spines before and during stimulation showed no accumulation of Alix at the postsynaptic level (**S1B and S1C Fig**). Instead, quantification of the distance between Alix signals and synapsin or PSD95 and analysis of colocalization supported the conclusion that synaptic activation leads to Alix recruitment at presynaptic parts only (**Fig 1I and 1J**).

### Recruitment of Alix partners, ALG-2 and endophilin-A, at activated synapses

In non-neuronal cells, Alix recruitment to plasma membrane depends on calcium binding to ALG-2, a penta-EF-hand-containing protein, which thereafter interacts with and activates Alix [13]. In neurons, action potential depolarization induces massive and transient calcium accumulation in the bouton, which triggers fusion of SV with the plasma membrane. Adding the intracellular calcium chelators BAPTA-AM or EGTA-AM completely abolished Alix-YFP recruitment (**S2 Fig**). Interestingly, GFP-ALG-2 concentrated at presynaptic boutons upon synaptic activation (**Fig 2A and 2B**). In contrast, an ALG-2 point-mutant unable to bind calcium (ALG-2ΔCa) [26] was not recruited upon activation with Bic/4AP showing that ALG-2 recruitment strictly depends on its capacity to bind calcium (**Fig 2B and 2C**). The use of neurons prepared from Alix ko mice [6,8] showed that ALG-2 recruitment does not depend on Alix, as it concentrated at stimulated synapses of Alix ko neurons at the same level as in wt neurons (**Fig 2B**). On the contrary, Alix recruitment to active synapses is tightly dependent on

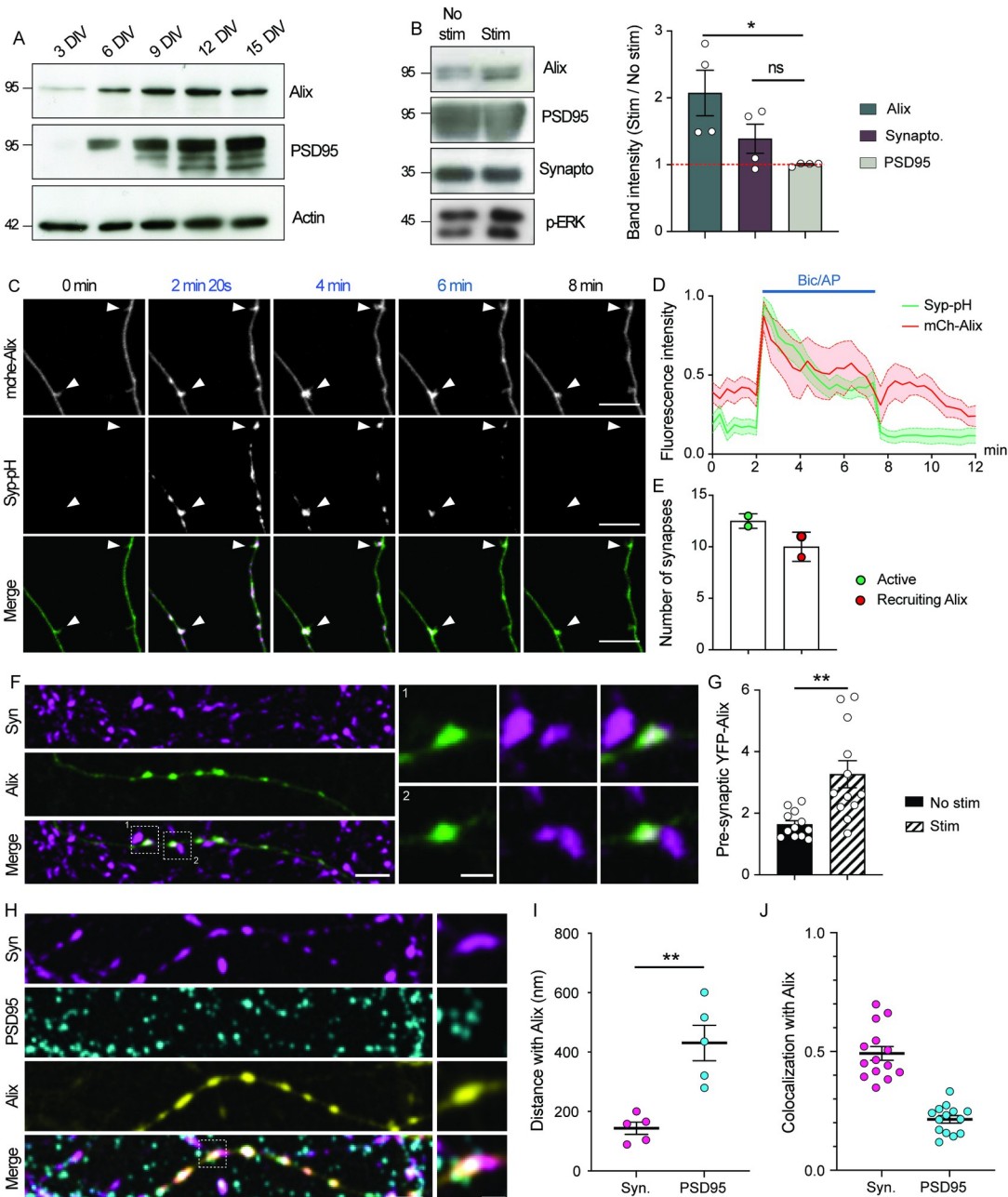

**Fig 1. Alix is recruited presynaptically during synaptic activation.** (**A**) Western blot analysis of cortical neurons cultured for 3 to 15 DIV demonstrates the increase in Alix expression correlating with synaptogenesis as illustrated by the increase in PSD95 expression. (**B**) Western blot analysis of the increase of Alix in synaptosome-enriched neuronal membranes upon neuronal stimulation by Bic/4AP. Synaptophysin and PSD95 were used as pre- and postsynaptic markers, respectively. The phosphorylated form of ERK (p-ERK) assessed the efficiency of the stimulation. (**C**) Images from time-lapse video microscopy of 15 DIV hippocampal neurons expressing both mche-Alix and Syp-pH stimulated between 2 and 7 min with Bic/4AP. White arrowheads indicate presynaptic boutons where Alix is recruited during stimulation. Scale bar: 10 μm. (**D**) mche-Alix and Syp-pH fluorescence variation at presynaptic boutons during Bic/4AP incubation (blue line). (**E**) Number of synapses responding to Bic/AP stimulation (Syp-pH increase) and recruiting Alix (mche-Alix increase). (**F**) 15 DIV hippocampal neurons expressing Alix-YFP (green) were stimulated for 5 min with Bic/4AP before fixation and stained with anti-synapsin-1 antibody (Syn, magenta). Dashed white squares indicate insets 1 and 2. Scale bars: 5 and 1 μm (inset). (**G**) Graph shows the presynaptic increase in Alix-YFP upon stimulation. Presynaptic Alix-YFP corresponds to the ratio of YFP fluorescence between synapsin-positive and synapsin-negative axonal regions. (**H**) Selective recruitment of Alix to the presynaptic part on synapses: 15 DIV hippocampal neurons expressing Alix-YFP (yellow) were stimulated for 5 min before fixation and stained with anti-synapsin-1 antibody (Syn, magenta) and anti-PSD95 (postsynaptic, cyan). Dashed white square indicates the inset on a single synaptic

bouton. Scale bars: 5 and 1 μm (inset). (I) Quantification of the distance between the peak of fluorescence between Alix and synapsin (magenta) or Alix and PSD95 (cyan) shows that Alix is significantly closer to synapsin than to PSD95. (J) Colocalization coefficient (Pearson's correlation coefficient) showing a higher level of colocalization between Alix and synapsin (magenta) than that between Alix and PSD95 (cyan). Average +/− SEM, N, statistical analysis: (B) 2.07 +/− 0.34; 1.38 +/− 0.21; 1.00 +/− 0.01 for Alix, synaptophysin, and PSD95, respectively. N = 4 independent experiments, Alix versus PSD95, p = 0.0187, 1-way ANOVA. (D) N = 25 and 16 synapses for Syp-pH and mChe-Alix, respectively, from 2 independent experiments. (E) Active synapses: 12.5 +/− 0.7; recruiting Alix: 10 +/− 1.4. N = 25 synapses from 2 independent experiments. (G) 1.63 +/− 0.12; 3.27 +/− 0.44 for no stim and stim, respectively. N = 12 neurons per condition from 4 independent experiments, p = 0.0017, unpaired t test. (I) Syn: 143.6 +/− 46 nm, PSD95: 430.9 +/− 133.7 nm. N = 5 independent experiments (40 synapses). Syn vs. PSD95, p = 0.0019, unpaired t test. (J) Syn: 0.49 +/− 0.1, PSD95: 0.21 +/− 0.06. N = 14 neurons from 5 independent experiments. The data underlying all the graphs shown in the figure can be found in S1 Data. Alix, ALG-2-interacting protein X; DIV, day *in vitro*; mche-Alix, mCherry-Alix; PSD95, postsynaptic density protein 95; Syp-pH, synaptophysin-pHluorin.

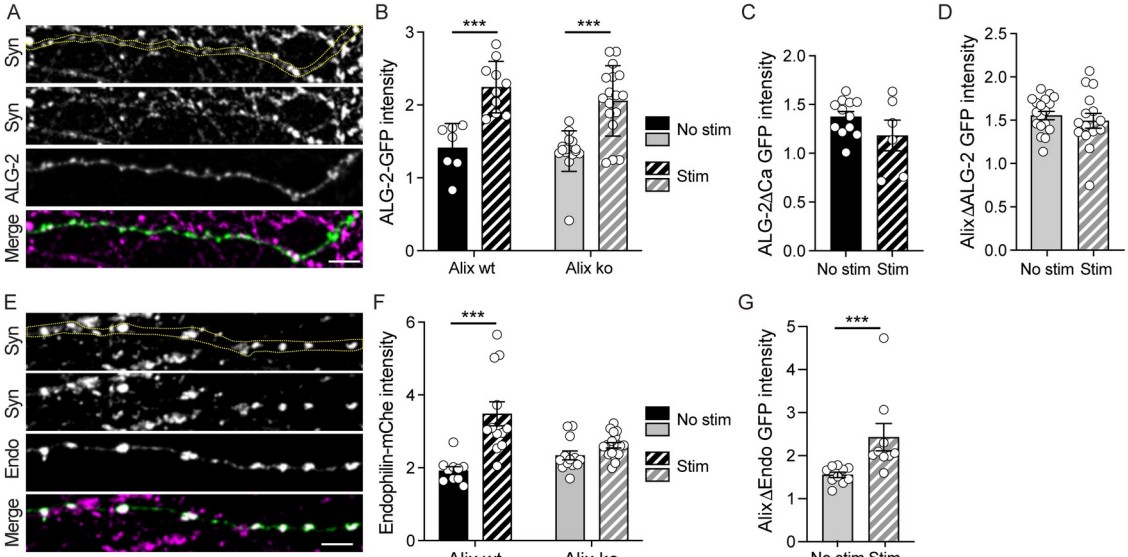

**Fig 2. Interplay between Alix, ALG-2, and endophilin recruitments at activated synapses.** All experiments made use of 15 DIV hippocampal neurons expressing the indicated constructs. Neurons were stimulated with Bic/4AP for 5 min before fixation and stained with anti-synapsin-1 (Syn, magenta). (**A**) GFP-ALG-2 (green) is recruited presynaptically upon stimulation. The yellow dashed line indicates the position of the axon in the synapsin-I channel (upper panel). Scale bar: 5 μm. (**B**) Quantification of synaptic recruitments of GFP-ALG-2 in Alix wt (black) and Alix ko (gray) neurons shows that ALG-2 presynaptic recruitment does not depend on Alix. (**C**) Quantification of synaptic recruitments of GFP-ALG-2ΔCa in Alix wt neurons (black) shows that ALG-2 presynaptic increase depends on its capacity to bind calcium. (**D**) Quantification of synaptic recruitments of GFP-AlixΔALG-2 in Alix ko neurons shows that Alix recruitment depends on its capacity to bind ALG-2 as GFP-AlixΔALG-2 is not recruited upon stimulation of Alix ko neurons. (**E**) Endophilin-mCherry (endo, green) concentrates at presynaptic parts of stimulated neurons. The yellow dashed line indicates the position of the axon in the synapsin-I channel (upper panel). Scale bar: 5 μm. (**F**) Synaptic recruitment of endophilin-mCherry following synaptic activation requires Alix as it is impaired in Alix ko neurons. (**G**) Quantification of synaptic recruitments of GFP-AlixΔendo in Alix ko neurons shows that Alix recruitment does not depend on its binding to endophilins. Average +/− SEM, N, statistical analysis: (B) 1.41 +/− 0.13; 2.24 +/− 0.11; 1.36 +/− 0.06; 2.06 +/− 0.12 for Alix wt no stim, Alix wt stim, Alix ko no stim, Alix ko stim, respectively. N = 7, 10, 17, 17 neurons for GFP-ALG-2 in Alix wt no stim, Alix wt stim, Alix ko no stim, Alix ko stim, respectively, from 3 experiments. p = 0.0001 for all the conditions tested, 1-way ANOVA. (C) 1.8 +/− 0.05; 1.18 +/− 0.16 for GFP-ALG2ΔCa in no stim and stim conditions, respectively. N = 12 and 6 neurons for no stim and stim, respectively. (D) 1.54 +/− 0.05; 1.48 +/− 0.08 for GFP-AlixΔALG-2 in no stim and stim conditions, respectively. N = 18 and 15 neurons for no stim and stim, respectively, from 3 independent experiments. (F) 1.92 +/− 0.11; 3.48 +/− 0.33; 2.34 +/− 0.12; 2.61 +/− 0.08 for endophilin-mche in Alix wt no stim, Alix wt stim, Alix ko no stim, Alix ko stim, respectively. N = 10, 12, 13, 19 neurons for Alix wt no stim, Alix wt stim, Alix ko no stim, Alix ko stim, respectively, from 3 independent experiments. Alix wt no stim vs. Alix wt stim, p < 0.0001, 1-way ANOVA. (G) 1.56 +/− 0.06; 2.43 +/− 0.32 for GFP-AlixΔendo in no stim and stim conditions, respectively. N = 10 and 9 neurons for no stim and stim, respectively, from 3 independent experiments. No stim vs. stim, p = 0.0004, Mann–Whitney test. The data underlying all the graphs shown in the figure can be found in S1 Data. Alix, ALG-2-interacting protein X; DIV, day *in vitro*; ko, knockout; wt, wild type.

its capacity to interact with ALG-2, as a mutated version of the protein unable to interact with ALG-2 (AlixΔALG-2) [27,28] does not accumulate at presynaptic boutons upon stimulation (**Fig 2D**). Thus, these results suggest that the calcium rise following synaptic activation allows calcium binding to ALG-2 that can interact with the membrane and activate Alix, a necessary step for Alix recruitment.

Among demonstrated interactors of Alix, endophilins-A are main regulators of endocytosis at synapses and impact the number of SVs [29,30]. We therefore tested if endophilins-A could be recruited to active synapses, similarly to Alix and ALG-2. Endophilin-A2-mCherry was mainly detected at presynaptic boutons and its fluorescence increased during Bic/4AP treatment (**Fig 2E and 2F** and **S2 Movie**). Remarkably, no such increase could be seen in Alix ko neurons (**Fig 2F**), whereas Alix deleted of its endophilin-binding domain (AlixΔendo) [11], was still able to be recruited to synapses upon activation (**Fig 2G**). This strongly suggests that the increase in endophilin concentration during sustained synaptic activation requires Alix expression.

## Alix is required for ADBE

Altogether, our observations indicate that calcium increase at synapses undergoing sustained activity allows the transient recruitment of ALG-2, Alix, and endophilin-A sequentially to presynaptic parts. Because of the demonstrated roles of Alix in CIE, we next tested if the turnover of presynaptic vesicles might be affected by the lack of the protein by quantifying Syp-pH fluorescence during electrical stimulation. Syp-pH fluorescence increases during stimulation witnessing exocytosis. It decays thereafter reflecting SV retrieval and vesicle re-acidification [31]. As shown in **Fig 3A–3D**, no striking difference in the increase of fluorescence was detected between Alix ko and wt neurons stimulated at 5 or 40 Hz (see also **S3A and S3B Fig**). However, the time constant of the exponential decay was slightly decreased in Alix ko neurons stimulated at 40 Hz, suggesting a higher rate of Syp-pH endocytosis occurring at this frequency (**Fig 3C and 3D**).

We next tested if the proteins might intervene in ADBE at synapses. Cultured cerebellar granular neurons (CGNs) have been extensively used to study this process [32]. Here, neurons were depolarized with 50 mM KCl, in the presence of horseradish peroxidase (HRP) that is endocytosed and fills vesicles and endosomes as they form. Examination of wt synapses by electron microscopy showed that depolarization dramatically increased the number of HRP-positive vacuoles. We verified using focused ion beam-scanning electron microscopy (FIB-SEM) and 3D reconstruction that these vacuoles had undergone fission from the plasma membrane and could therefore be identified as bulk endosomes (**Fig 3E–3G** red arrowheads and **S3 Movie**). Other HRP-positive vesicles having the size of neurotransmitter vesicles were also more numerous in depolarized synapses (**Fig 3E–3G** blue arrowheads). In Alix ko synapses, the number of depolarization-induced bulk endosomes was strongly reduced (**Fig 3G and 3I**). Consistent with the increased endocytic rate detected by Syp-pH in Alix ko neurons stimulated at 40 Hz (**Fig 3C and 3D**), the number of newly formed SVs was slightly but significantly increased, suggesting a mechanism compensating for ADBE deficiency (**Fig 3G and 3J**). Interestingly, bulk endosomes formed in Alix ko synapses were significantly smaller than in Alix wt synapses (**Figs 3H and 3K** and **S3C–S3E**).

Another way to assess for ADBE is by the use of fluorescent 10 kDa dextran, a fluid phase cargo that accumulates inside bulk endosomes but fails to label vesicles formed by clathrin-dependent mechanisms upon neuronal stimulation [33]. Hippocampal neurons incorporated dextran when stimulated with Bic/4AP (**Fig 4A and 4B**). Similar results were obtained upon depolarization with 50 mM KCl (**S4A and S4B Fig**). A GSK3 inhibitor, known to block ADBE

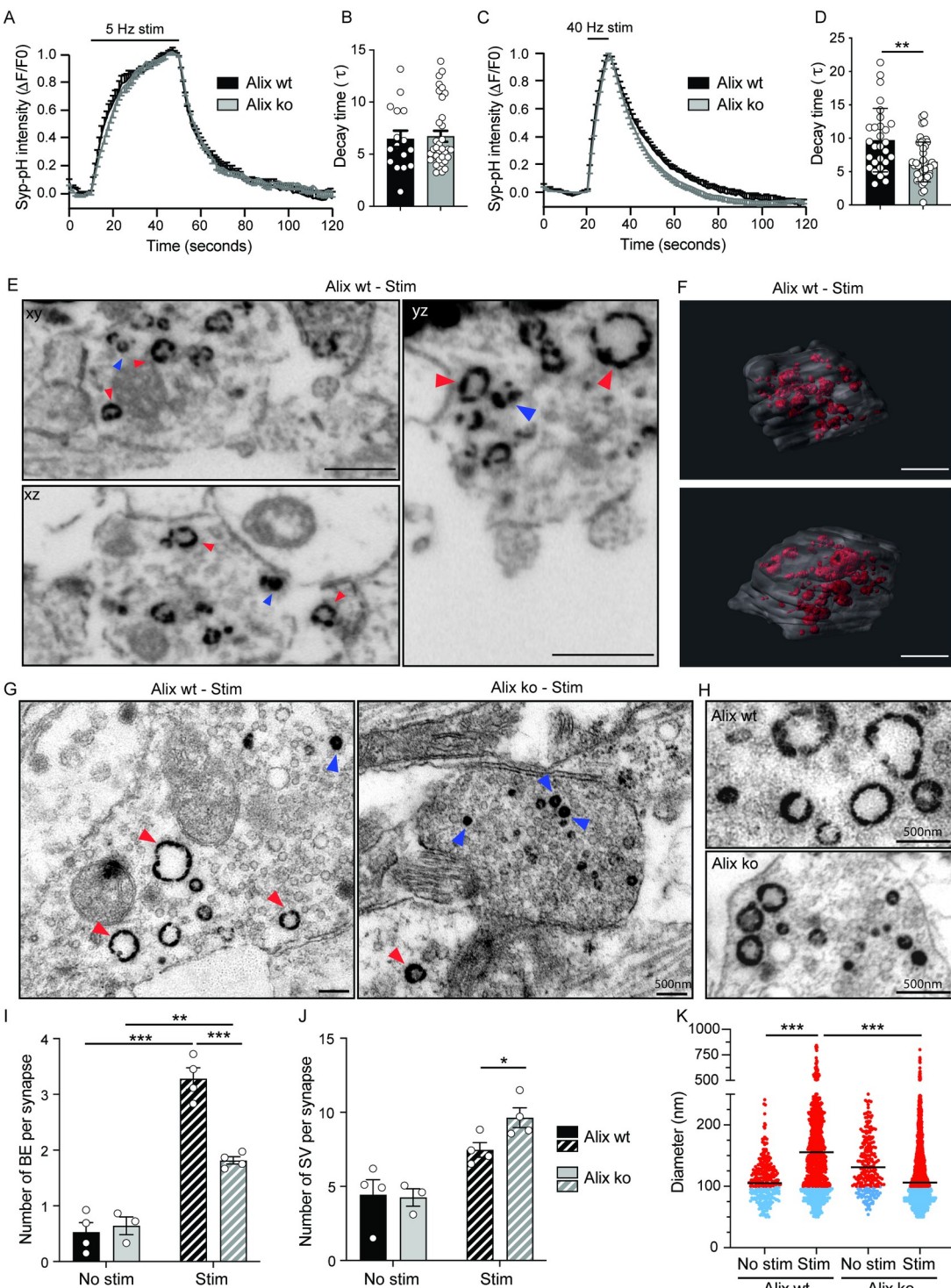

**Fig 3. Alix is necessary for activity-dependent bulk endocytosis.** (**A**, **C**) Normalized traces of Syp-pH fluorescence in synaptic boutons of Alix wt and ko hippocampal neurons stimulated with 200 APs applied at 5 Hz (A) or 40 Hz (C). (**B**, **D**) The exponential fit of fluorescence decay in the imaged fields after stimulations show an increased decay in Alix ko neurons at 40 Hz (D) but not at 5 Hz (B). (**E**, **H**) Electron micrographs of cerebellar granule neurons stimulated in presence of free HRP to label newly formed SV (blue arrowheads) and bulk endosomes (red arrowheads). (**E**) FIB-SEM orthogonal views from different planes (xy, xz, yz) extracted from a stack used for the 3D reconstruction of a wt presynaptic terminal shown in (F). Scale bar: 500 nm. (**F**) Two different views of the reconstructed synapse are shown where the membrane is represented in transparent gray and HRP-positive structures in red. Scale bar: 500 nm. (**G, H**) Transmission electron microscopy images of Alix wt and Alix ko

cerebellar granule neurons incubated with HRP. Scale bars: 500 nm. (**I, J**) Quantification of the number of bulk endosomes (I) and SVs (J) in Alix wt and ko neurons from images as shown in (G). (**K**) Quantification of the diameter of SV (blue dots) and bulk endosome (red dots) in Alix wt and Alix ko neurons in basal or stimulated condition showing that bulk endosomes of Alix ko synapses are smaller than in Alix wt neurons. <u>Average +/− SEM, N, statistical analysis:</u> (B, D) 6.47 +/− 0.78; 6.72 +/− 0.53; 9.71 +/− 0.88; 6.45 +/− 0.44 for Alix wt 5 Hz, Alix ko 5 Hz, Alix wt 40 Hz, Alix ko 40 Hz, respectively. $N$ = 16, 35, 29, 46 fields of view per condition from 4 experiments for wt and 5 for ko mice. Alix wt vs. Alix ko, $p$ = 0.99 (B) and $p$ = 0.002 (D), Mann–Whitney test. (I) 0.52 +/− 0.17; 0.64 +/− 0.16; 3.28 +/− 0.19; 1.82 +/− 0.07 for Alix wt no stim, Alix ko no stim, Alix wt stim, Alix ko stim, respectively. $N$ = 4, 3, 4, 4 independent experiments Alix wt no stim, Alix ko no stim, Alix wt stim, Alix ko stim, respectively. Alix wt no stim vs. Alix wt stim, $p$ < 0.0001; Alix ko no stim vs. Alix ko stim, $p$ = 0.0015; Alix wt stim vs. Alix ko stim, $p$ = 0.0001, 1-way ANOVA. (J) 4.43 +/− 1.02; 4.26 +/− 0.59; 7.46 +/− 0.49; 9.64 +/− 0.67 for Alix wt no stim, Alix ko no stim, Alix wt stim, Alix ko stim, respectively. $N$ = 4, 3, 4, 4 independent experiments Alix wt no stim, Alix ko no stim, Alix wt stim, Alix ko stim, respectively. Alix wt stim vs. Alix ko stim, $p$ = 0.039, unpaired $t$ test. (K) 112.2 +/− 2.37 nm; 191.7 +/− 3.39 nm; 152.4 +/− 4.1 nm; 135.2 +/− 1.79 nm for Alix wt no stim, Alix wt stim, Alix ko no stim, Alix ko stim, respectively. $N$ = 304, 1,326, 285, 2,415 vesicles for Alix wt no stim, Alix wt stim, Alix ko no stim, Alix ko stim, respectively, from 3 independent experiments. $p$ < 0.0001 in all conditions tested, Kruskal–Wallis test. The data underlying all the graphs shown in the figure can be found in S1 Data. Alix, ALG-2-interacting protein X; AP, action potential; FIB-SEM, focused ion beam-scanning electron microscopy; HRP, horseradish peroxidase; ko, knockout; SV, synaptic vesicle; Syp-pH, synaptophysin-pHluorin; wt, wild type.

but no other modes of SV endocytosis [34], completely abolished the dextran labeling of wt hippocampal neurons stimulated with Bic/4AP (**S4C and S4D Fig**). In contrast, Alix ko neurons incubated in presence of Bic/4AP failed to endocytose dextran (**Fig 4A and 4B**) even

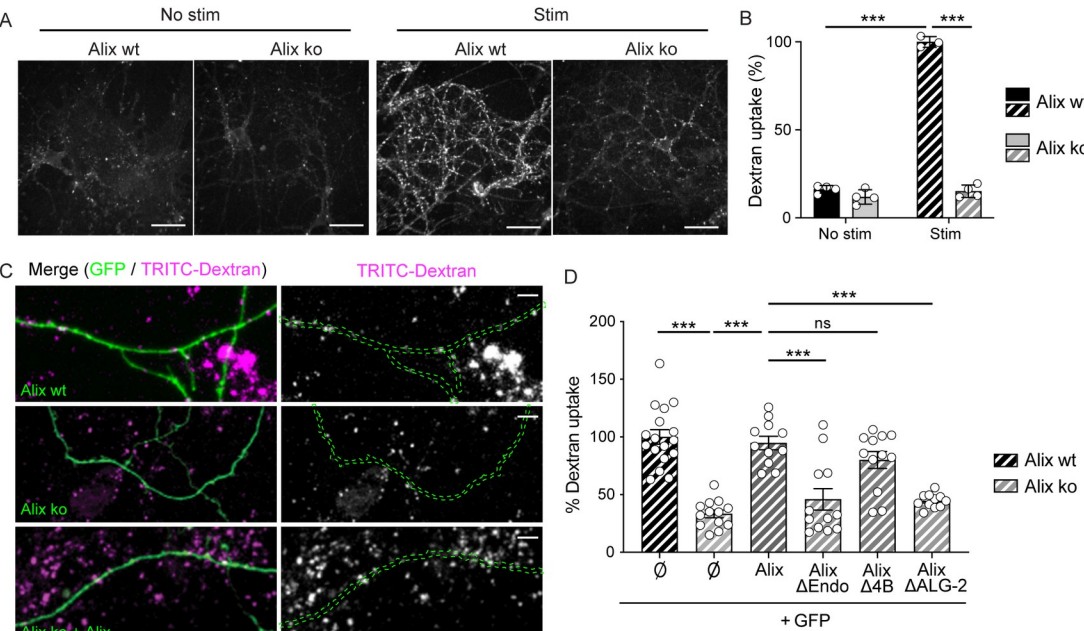

**Fig 4. Alix-driven bulk endocytosis requires its binding to ALG-2 and endophilin but not to CHMP4.** (**A**) Confocal images of Alix wt and Alix ko hippocampal neurons stimulated with Bic/4AP in the presence of 10 kDa dextran. Scale bar: 50 μm. (**B**) Dextran uptake triggered by stimulation is strongly reduced in Alix ko neurons. (**C**) Representative images of dextran uptake by GFP-expressing Alix wt and Alix ko neurons (Alix wt, top and Alix ko, middle) or Alix ko neurons expressing both GFP and Alix (Alix ko + Alix, bottom). Scale bars: 10 μm. (**D**) Dextran uptake is rescued in ko neurons expressing Alix wt and AlixΔChmp4B (AlixΔ4B), but not AlixΔALG-2 or AlixΔendo. The % of dextran uptake corresponds to the number of dextran spots per μm expressed as percentages of the control (Alix wt) for each experiment. <u>Average +/− SEM, N, statistical analysis:</u> (B) 16.26 +/− 1.08; 11.86 +/− 2.05; 100 +/− 1.76; 15.10 +/− 1.76 for Alix wt no stim, Alix ko no stim, Alix wt stim, Alix ko stim, respectively. $N$ = 4 experiments, $p$ = 0.0001, 1-way ANOVA. (D) 100 +/− 6.31; 33.27 +/− 3.32; 94.78 +/− 5.81; 45.89 +/− 9.36; 80.05 +/− 7.32; 44.59 +/− 1.85 for Alix wt, Alix ko, Alix ko + Alix, Alix ko + AlixΔChmp4B, Alix ko + AlixΔendo, Alix ko + AlixΔALG-2, respectively. $N$ = 17 neurons for Alix wt, $N$ = 13 neurons for Alix ko, $N$ = 11 for Alix ko + Alix, $N$ = 12 for Alix ko + AlixΔChmp4B, $N$ = 12 neurons for Alix ko + AlixΔendo, and $N$ = 11 for Alix ko + AlixΔALG-2, all from 3–4 independent experiments, $p$ = 0.62 (Alix wt vs. Alix AlixΔ4B), $p$ = 0.0001 for the other conditions, 1-way ANOVA. The data underlying all the graphs shown in the figure can be found in S1 Data. Alix, ALG-2-interacting protein X; ko, knockout; wt, wild type.

though stimulation increased calcium entry (**S1A Fig**) and neuronal activity (**S4E and S4F Fig**) in both Alix ko and wt neurons. A similar impairment in dextran endocytosis by Alix ko neurons was revealed upon depolarization with 50 mM KCl (**S4A and S4B Fig**). As already shown [35], neurons also failed to endocytose dextran when treated with calcium chelators BAPTA and EGTA (**S4G Fig**), both of which blocked Alix recruitment to presynaptic boutons (**S2 Fig**). Finally, expression of GFP-Alix constructs in Alix ko neurons showed that the impairment in ADBE observed in Alix ko cells is due solely to the absence of Alix, since Alix expression fully restored the capacity of Alix ko neurons to endocytose 10 kDa dextran (**Fig 4C and 4D**). Rescue was also obtained with AlixΔ4B, a mutant unable to bind the ESCRT-III protein CHMP4B. This was not the case with AlixΔALG-2 or AlixΔendo that were not able to rescue dextran endocytosis (**Fig 4D**). Noteworthy, AlixΔALG-2 cannot be recruited to synapses but both AlixΔendo and AlixΔ4B are. Thus, besides its requirement for ALG-2-binding to be recruited at synapses, Alix needs to interact with endophilins but not with CHMP4B of ESCRT-III in order to drive ADBE.

## Synaptic anomalies detected in Alix ko mice

To study the effect of Alix deletion *in vivo*, we first compared brains of Alix ko [8] and wt mice and found that the density of synaptic contact as well as dendritic spines in the CA1 hippocampal region of adult brains was not different (**S5A and S5C Fig**). This indicates that Alix is not required for synaptogenesis, a conclusion confirmed *in vitro* since the number of synapses revealed by co-immunostaining with synapsin-1 and PSD95 was similar between wt and Alix ko cultured hippocampal neurons (**S5B Fig**). However, examination of the CA1 *stratum radiatum* of adult mice showed that Alix ko synapses contained significantly fewer SV than wt (**S6A and S6B Fig**). Furthermore, the size of postsynaptic densities, known to be strictly correlated with the number of SV [36], was similarly reduced in Alix ko synapses (**S6D and S6E Fig**). Importantly, the surface of Alix ko synaptic boutons was also significantly increased possibly suggesting plasma membrane accumulation in these synapses (**S6A and S6C Fig**). Finally, at the postsynaptic level, we also noticed that the ratio between the diameter of the spine head and that of the neck was changed in Alix ko animals, which can reflect defects in maturation or plasticity of adult synapses (**S6D and S6F Fig**). Accordingly, field recordings in the CA1 *stratum radiatum* of hippocampal slices revealed that long-term potentiation (LTP) induced by tetanic stimulations was significantly impaired in Alix ko hippocampal slices (**S6G Fig**). The impairment in LTP might reflect presynaptic defects of ADBE in Alix ko synapses. Further investigations are required to sustain this hypothesis and to verify that postsynaptic events underlying LTP are not impaired in Alix ko mice. This includes other LTP experimental paradigms that do not require presynaptic high frequency stimulation.

## Voltage clamp recordings uncover anomalies at Alix ko synapses

In order to better characterize the effect of the lack of Alix on the physiology of synapses, we next used Alix conditional ko (cko) mice where Alix is deleted in neocortical and hippocampal excitatory neurons [37] (**Fig 5A**). We first recorded CA1 neurons of hippocampal slices and found that the amplitude of spontaneous excitatory postsynaptic currents (sEPSCs) was unaffected, whereas the frequency of events was significantly lower in Alix ko neurons (**Fig 5B and 5C**). This was reminiscent of the reduced frequency of bursts measured by multielectrode arrays (MEAs) in dissociated hippocampal cultures (**S4E and S4F Fig**). Stimulation of Schaffer collaterals revealed that evoked excitatory postsynaptic currents (EPSCs) have a smaller amplitude in Alix ko brains than in wt animals (**Fig 5D and 5E**). To estimate the amount of the readily releasable pool (RRP), we performed repeated stimulations and integrated the EPSC

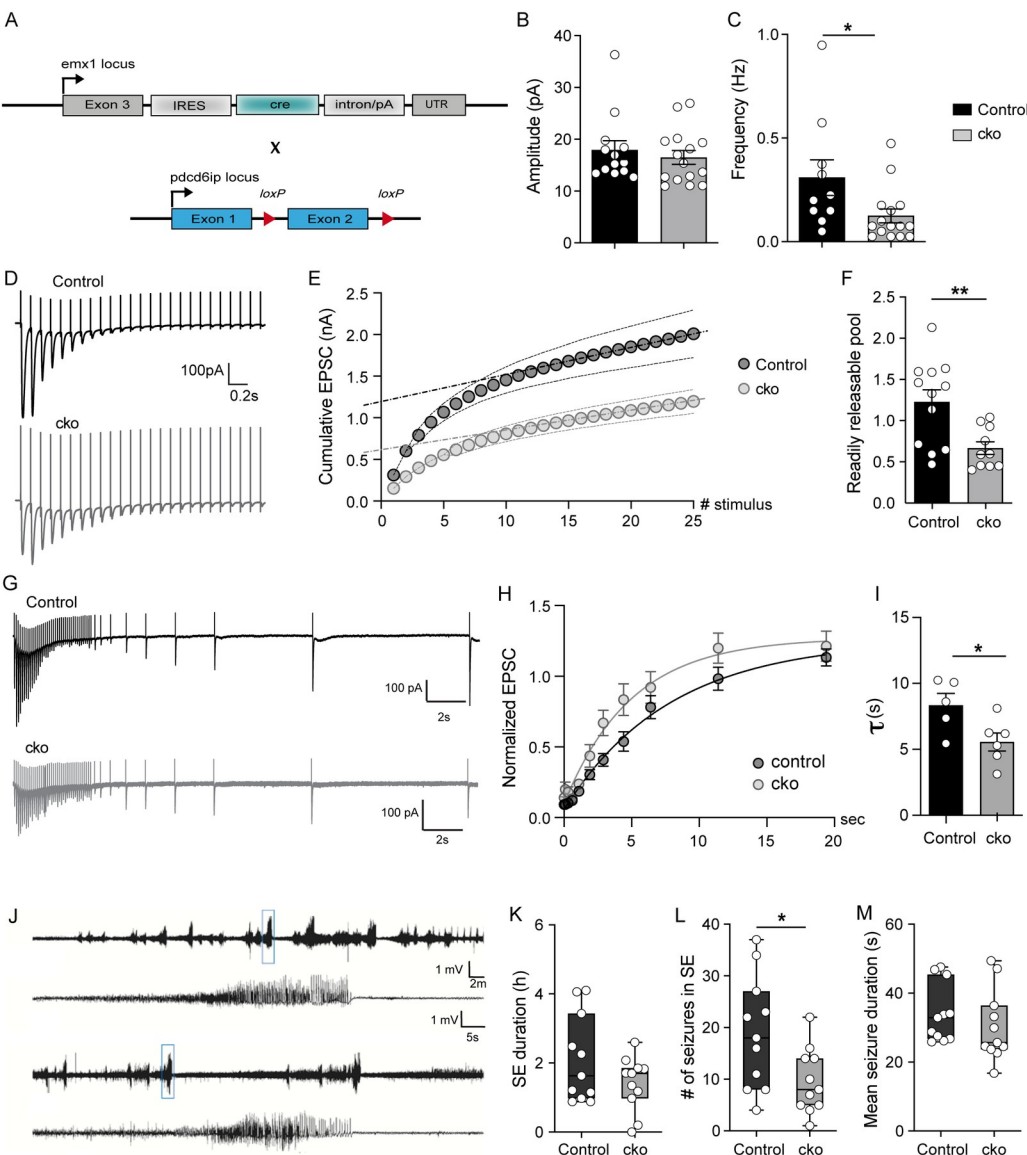

**Fig 5. Alterations of Alix ko synapses revealed in hippocampal slices and in an *in vivo* model of epilepsy.** (**A**) Emx1 IREScre (Emx-Cre) and Alix fl/fl mouse lines were crossbred to delete Alix in neocortical and hippocampal excitatory neurons (Alix cko). (**B**) No difference in sEPSC amplitude was detected between control and cko mice. (**C**) The frequency of sEPSC in cko neurons is lower compared to control. (**D**) Representative traces showing short-term depression in response to 10 Hz stimulation trains in control (black) and cko (gray) mice. (**E**) Cumulative EPSC amplitudes in response to 10 Hz stimulation. Train-extrapolation is illustrated by the dashed line. (**F**) The size of the RRP, estimated by the train-extrapolation method, was significantly decreased in Alix cko mice. (**G**) Representative traces showing recovery from depression in response to 10 Hz stimulation trains in control (black) and Alix cko (gray) mice. (**H**) To evaluate the speed of recovery, all inward signals were normalized to the first inward current. Recovering signals with a single-exponential function revealed the plateaus at which the capacities of recovery were saturated. (**I**) Alix cko neurons recover slightly faster than controls. (**J**) Representative EEG traces from control and Alix cko mice. (**K**) Total duration of SE did not differ between Alix cko and control mice. (**L**) Alix cko mice experience about 66% fewer seizures during SE than controls. (**M**) Mean seizure duration during SE was not affected in Alix cko mice compared to controls. <u>Average +/− SEM, *N*, statistical analysis:</u> (B) 17.93 +/− 1.81; 16.49 +/− 1.35 for controls and Alix cko, respectively. *N* = 13 neurons from 5 control mice and *N* = 15 neurons from 6 Alix cko mice. (C) 0.31 +/− 0.09; 0.12 +/− 0.03 for controls and Alix cko, respectively. *N* = 10 neurons from 5 control mice and *N* = 15 neurons from 6 Alix cko mice. Control vs. cko, *p* = 0.015, Mann–Whitney test. (F) 1.23 +/− 0.15; 0.67 +/− 0:08 for controls and Alix cko, respectively. *N* = 12 neurons from 5 control mice and *N* = 10 neurons from 4 Alix cko mice. Control vs. cko, *p* = 0.0052, unpaired *t* test. (I) 8.34 +/− 0.90; 5.56 +/− 0.69 for controls and Alix cko, respectively. *N* = 5 and 6 neurons from 3 mice from control and Alix cko, respectively. Control vs. cko, *p* = 0.0336, unpaired *t* test. <u>Median (IQR), *N*, statistical analysis:</u> (K) Control: 1.6 h (IQR = 2.5); cko: 1.7 h (IQR = 0.9).

$N$ = 11 mice for both genotypes. (L) Control: 18 seizures (IQR = 9); cko: 8 seizures (IQR = 9). $N$ = 11 mice for both genotypes. Control vs. cko, $p$ = 0.023, Mann–Whitney test. (M) Control: 32.9 s (IQR = 18.6); cko: 25.7 h (IQR = 12.7). $N$ = 11 mice for both genotypes. The data underlying all the graphs shown in the figure can be found in S1 Data. Alix, ALG-2-interacting protein X; cko, conditional ko; IQR, interquartile range; RRP, readily releasable pool; SE, status epilepticus; sEPSC, spontaneous excitatory postsynaptic current.

amplitudes to give a cumulative plot during 10 Hz trains. Linear regression fit to the last 10 data points was back-extrapolated to time 0 (dotted line) to estimate the cumulative EPSC before steady-state depression. This rough estimate of the RRP size [38] revealed that it is halved in Alix ko neurons (**Fig 5E and 5F**), in good agreement with the lower numbers of SVs quantified in EM sections of synapses of the CA1 *stratum radiatum* (**S6B Fig**). Interestingly, we could observe that the rate of recovery postdepression appears faster in Alix ko mice, a possible mechanism of compensation for the reduced pool of SVs (**Fig 5G–5I**).

Thus, these electrophysiological results show that synapses of Alix ko neurons which lack ADBE have fewer SVs but increased rate of endocytosis, as already suggested in cultured neurons by the increase of Syp-pH endocytosis and of newly formed SVs detected in Alix ko-hippocampal and CGN, respectively.

## Alix conditional ko mice undergo fewer kainate-induced acute seizures

We next examined the possible *in vivo* consequences of synaptic changes detected during repeated stimulation of Alix ko neurons in hippocampal slices. Because of our earlier finding of Alix increase at synapses of the rat hippocampus after kainate injections [9], we tested how the lack of Alix might influence kainate-induced epileptic seizures, a mouse model for human temporal lobe epilepsy [39]. The unilateral intracortical kainate injection induces the acute phase, SE, characterized by high seizure frequency, during several hours. Seizure activity was determined by telemetric electroencephalogram (EEG) recording and synchronized video monitoring. Here, we used the cko mice where Alix is deleted in neocortical and hippocampal excitatory neurons (**Fig 5J**). The total duration of SE was comparable in Alix cko and control mice (**Fig 5K**). However, Alix cko mice experienced about 70% fewer seizures than control mice (**Fig 5L**). The average seizure duration was only minimally reduced in Alix cko compared to control (**Fig 5M**). These results show that the absence of Alix selectively reduces the number of high frequency events (single seizures), without affecting the total duration of SE.

This kainate injection paradigm produces seizures originating from the hippocampus, which can spread to other contralateral brain areas [39]. Microglial reactivity was detected 24 h after kainate injection in ipsilateral and contralateral cortex and hippocampus in both control and Alix cko animals (**S7A–S7E Fig**). While the difference between ipsi- and contralateral side was non significant in control animals (≈10%), Alix ko animals displayed a significantly reduced contralateral microglial activation (≈30%) (**S7E Fig**). No difference between Alix cko and control animals was detected using glial fibrillary acidic protein (GFAP) immunolabeling, an expected result since astroglial activation peaks 4 to 7 days after kainate injection (**S7F Fig**). Thus, the reduced contralateral microglial activation seen in Alix cko animals indicates a reduced propagation of the seizure activity and corroborates the previous seizure quantification (**Fig 5L**).

In summary, Alix is specifically required for ADBE. Impairment in this process seems to relate to impairments in synaptic function and plasticity in normal and pathological settings.

## Discussion

To sustain neurotransmission and prevent expansion of the presynaptic plasma membrane, SV fusion is coupled to the endocytic recycling and regeneration of SV proteins and lipids.

Vesicle components can be retrieved from the plasma membrane via clathrin scaffolds, or via clathrin-independent processes mediating fast and ultrafast endocytosis and, in the case of high frequency stimulation, bulk endocytosis [1]. Using Alix ko cells, we previously reported that Alix drives CIE during ligand-induced endophilin-dependent endocytosis as well as bulk endocytosis [6,8]. In another set of experiments, we also showed that Alix immunoreactivity in the rat hippocampus is strongly up-regulated in synapses undergoing high frequency activation during kainate-induced epileptic seizures. This increase was presynaptic and only transient, as it was reversed soon after cessation of the seizures [9]. The aim of the present study was to decipher the physiological role of Alix in endocytosis at synapses. We first used primary cultures of cortical and hippocampal neurons, which make networks undergoing spontaneous activities. Massive increases in spike frequency induced by 4AP and bicuculline resulted in elevated calcium entry [23] and triggered ADBE together with a transient concentration of Alix at presynaptic parts. Quantifying bulk endocytosis using dextran uptake or electron microscopy observations showed that ADBE is strongly reduced in Alix ko neurons demonstrating that Alix is essential for ADBE. ADBE is triggered by high $[Ca^{2+}]$ in response to sustained activity [40]. Calcium and calmodulin-dependent phosphatase calcineurin were suggested to make a link between synaptic activity and formation of bulk endosomes through dynamin [35]. Alix may represent another link as its recruitment and activity in ADBE seems dependent on calcium increase. Indeed, EGTA, which binds calcium at an approximately 100 times slower on-rate than BAPTA [41,42] and was already known to block ADBE without affecting synaptic release, was equally efficient as BAPTA in inhibiting Alix recruitment. Furthermore, Alix recruitment required its capacity to bind the calcium-binding protein ALG-2, which also concentrated at hyperactive synapses. ALG-2 is a cytosolic penta-EF-hand-containing protein with 2 $Ca^{2+}$-binding sites (Kd = 1.2 μM), and its interaction with Alix strictly depends on calcium [10,26,27]. Conformational change and exposure of hydrophobic patches occur at μM concentrations of $Ca^{2+}$, suggesting that ALG-2 functions as a calcium sensor [43]. Indeed, in non-neuronal cells, calcium entry provoked by membrane wounds leads to the sequential recruitment to the membrane of ALG-2, Alix, and ESCRT-III proteins necessary for membrane repair [13,44]. In neurons, the activity-dependent accumulation of ALG-2 at synapses required $Ca^{2+}$ binding as it was totally abolished by point-mutations within the 2 $Ca^{2+}$-binding sites. On the other hand, it did not require Alix, as ALG-2 recruitment also occurred in Alix ko neuron. Thus, our observations highlight synaptic ALG-2 as an obvious candidate for sensing calcium elevation and suggest the following scenario: Sustained high frequency depolarization leads to massive elevation of calcium in the synaptic bouton; $Ca^{2+}$ binding to ALG-2 causes its accumulation at the synaptic membrane and binding to cytosolic Alix, whose recruitment to the membrane drives ADBE.

Little is known about the molecular mechanisms underlying the plasma membrane deformation during ADBE. The proline-rich domain of Alix interacts with the SH3 domains of endophilins-A1-3 that contain N-BAR domains and have been shown to regulate clathrin-dependent and -independent SV endocytosis at different synapses [16,17,20]. BAR domains are dimerization domains able to induce, stabilize, and sense membrane curvature [15,45]. Interestingly, a proteomic approach revealed the presence of endophilin-A1, Alix, and ALG-2 in bulk endosomes [46]. Furthermore, endophilin-A2, actin, and dynamin mediate a restricted type of CIE activated upon ligand binding to cargo receptors [18,19]. We previously showed that in fibroblasts, endophilin-A and Alix act in the same CIE pathway, even if Alix is more promiscuous for ligands, as it is also involved in fluid phase endocytosis. In this case, interaction of Alix with endophilins favored their presence at the membrane [6]. We now found that endophilin-A2 decorated presynapses of resting neurons and that Alix was needed for its synaptic enrichment during stimulation. An Alix point-mutant unable to interact with endophilin

was recruited to synapses but could not rescue ADBE in Alix ko neurons, reminiscent of the situation seen in fibroblasts [6]. Thus, a complex between endophilin-A and Alix is required for ADBE. One function of Alix might be to enable endophilin-A binding onto flat membranes of the peri-active zone to drive membrane bending in response to high calcium concentrations [15,18]. Interestingly, the incapacity of Alix to interact with Chmp4B from ESCRT-III, known to alleviate its plasma membrane repair ability [13,44], did not alter rescue of ADBE. This result also discriminates Alix-driven ADBE from roles of the ESCRT system in the degradation of SV proteins [47,48].

Electrophysiological recordings and EM observations in CA3-CA1 synapses both point to a reduction in the number of SVs due to the lack of Alix and possibly to the associated defects in ADBE. On the other hand, Syp-pH fluorescence and EM analysis revealed that activity-dependent synaptic endocytosis is increased in absence of Alix. Accordingly, electrophysiological patch-clamp recordings show that Alix ko CA3-CA1 synapses recover better from short-term depression (STD) induced by repeated stimulations. Thus, compensation by increased endocytosis is not sufficient to rescue the phenotype. This could be a first demonstration of the essential role of ADBE in regulating the number of SVs and thereby normal synaptic function.

Excitatory glutamatergic neurotransmission plays a central role in the generation of seizure activity [49,50] and figures among the primary antiepileptic drug targets [51]. Sustained, intense, and synchronous neurotransmission as occurring during seizures requires replenishment of SV pools mainly through ADBE [5]. Impairing ADBE *in vivo*, by deleting Alix in neocortical and hippocampal excitatory neurons led to a significantly reduced number of seizures during SE, without affecting the total duration of SE or the mean seizure duration. The present results suggest that excitatory networks lacking Alix have a reduced capacity to initiate sequential seizures, possibly due to an impaired replenishment of SVs through ADBE. Seizure termination and SE cessation however seem to be governed by distinct mechanisms, probably involving inhibitory transmission to re-establish the balance between excitation and inhibition in the brain. We would predict a role for Alix-driven ADBE across epilepsy types. The degree and nature of impact, however, might vary according to the etiology of the epileptic syndrome in question. In addition to our pharmacological kainate model, it would be interesting to analyze Alix-dependent seizure susceptibility also in genetic models with spontaneous seizures, such as mutations of the voltage-gated sodium channel SCN8A [52] or tuberous sclerosis complex 1/2 [53], as well as inbred mouse strains with high susceptibility to audiogenic seizures (DBA/1J, DBA/2J, or 129/SvTer strain) [54]. Our results highlight Alix ko mice as an invaluable tool for exploring and understanding the exact role of ADBE at synapses undergoing normal and pathological stimulations.

## Material and methods

### Plasmids

Endophilin A2-mCherry was obtained by subcloning (In-Fusion Cloning kit, 102518, Clontech) endophilin A2 cDNA into a pmCherry-N1 vector (Clontech). GFP-ALG-2 was obtained by performing a reverse mutagenesis (Quick change II site directed mutagenesis kit, 200523, Agilent) on a GFP-hALG-2 Y180A (a generous gift from Masatoshi Maki) to acquire GFP-hALG-2wt. hALG-2 E47A-E114A cDNA was kindly provided by Masatoshi Maki [26] and was subcloned into a pEGFP-C1 vector (Clontech) to obtain GFP-hALG-2 E47A-E114A (GFP-ALG-2ΔCa).

All constructs containing Alix cDNA (wt or mutants) were obtained by subcloning the relevant cDNAs from pCI vectors harboring Alix cDNA or its mutants. Alix I212D (AlixΔCHMP4B) and AlixΔPGY (AlixΔALG-2) cDNAs in pCI were generated by

mutagenesis (Quick change II site directed mutagenesis kit, 200523, Agilent) and Alix R757E (AlixΔendo) by in-fusion cloning, using the oligos given below.

mCherry-2Xflag-mAlix wt (mCherry-Alix) was obtained by subcloning 2xflag-mAlix wt cDNA into a pmCherry-C1 vector (Clontech). Alix-YFP was obtained by subcloning wild-type Alix cDNA into a pEYFP-N1 vector (Clontech). GFP-flag-Alix (GFP-Alix) was described in [6]. GFP-Alix and its mutant forms (GFP-Alix R757E, GFP-AlixΔPGY) were obtained by subcloning the various cDNAs into a pEGFP-C1 vector (Clontech). DNA constructs used for the rescue experiments were prepared in 2 steps. First, IRES2-GFP cDNA was subcloned into pSIN lentiviral vector (kindly provided by F. Saudou) by using pIRES2-GFP (Clontech) as a template. Then, the various cDNAs were subcloned into pSIN-IRES2-GFP.

Oligos used to generate mutants:

Alix I212D

> sense: 5′-AAGATGAAAG ATGCCGACAT AGCTAAGCTG-3′

> antisense: 5′-CAGCTTAGCT ATGTCGGCAT CTTTCATCTT-3′

Alix R757E

> sense: 5′-CAGCCGAGCC TCCACCTCCT GTGCTTCCTG-3′

> antisense: 5′-GAGGCTCGGC TGGAGGCTGG GGCTTAGCAG-3′

AlixΔPGY

> sense: 5′-GCCACAGGCT CAGGGATGCC AAATGCCCAT GC-3′

> antisense: 5′-GCATGGGCAT TTGGCATCCC TGAGCCTGTG GC-3′.

## Antibodies

See Table 1 for the list and information about the antibodies used in this study.

## Animals

Animals were handled and killed in conformity with European law and internal regulations of INSERM. Pregnant Oncins France souche A (OFA; a substrain of Sprague Dawley) rats (Charles River) were used for rat neuronal cultures. Alix ko C57BL/6 mouse pups [6,8] and their control littermates, referred to thereafter as Alix wt, were also used for primary neuronal culture. Transgenic mice were held at the animal facility of the Grenoble Institute for Neurosciences and fed *ad libitum*. All animals were held at a 12-h light/dark cycle. One- to 2-month-old Alix wt and Alix ko mice were used for electrophysiological extracellular recordings, histochemistry, and electron microscopy studies.

Mice were anesthetized by intraperitoneal injection of 0.1 ml sodium pentobarbital (5.6% w/v; CEVA Santé Animale) and treated as described in the corresponding subheadings of the Material and methods section.

Emx1$^{IREScre}$/Alix$^{fl/fl}$ and Emx1$^{IREScre}$ control mice, thereafter referred to as control and Alix cko, were used for the kainate injection epilepsy model and the electrophysiological. The 8 to 11 weeks old mice were anesthetized under a mixture of 2% isoflurane, 47.5% $O_2$, and 47.5% $N_2O$. Transgenic mice were held at the animal facility of the CIPMM and fed ad libitum. All animals were held at a 12-h light/dark cycle. This study was carried out in strict accordance with the European and German guidelines for the welfare of experimental animals. Animal

**Table 1. List of the antibodies used in this study.**

| Antibody | Supplier (reference) | Species (type) | Dilution |
|---|---|---|---|
| Anti-Actin | Millipore (MAB1501R) | Mouse (monoclonal) | 1/10,000 |
| Anti-Alix | Covalab (ab0204) | Rabbit (polyclonal) | 1/10,000 |
| Anti GFAP | Abcam | Goat (polyclonal) | 1/1,000 |
| Anti Iba1 | Wako | Rabbit (polyclonal) | 1/500 |
| Anti-phospho-p44/42 MAPK (T202/Y204) | Cell signaling (9106) | Mouse (monoclonal) | 1/1,000 |
| Anti-PSD95 | NeuroMab (73–028) | Mouse (monoclonal) | 1/2,000 |
| Anti-PSD95 | Millipore (MAB1598) | Mouse (monoclonal) | 1/500 |
| Anti-Synapsin-1 | Millipore (AB1543P) | Rabbit (polyclonal) | 1/1,000 |
| Anti-Synaptophysin | Merck Millipore (MAB5258) | Mouse (monoclonal) | 1/5,000 |
| Anti-Flag | Sigma-Aldrich (F3165) | Mouse (monoclonal) | 1/1,000 |
| Anti-Flag | Sigma-Aldrich (F7425) | Rabbit (polyclonal) | 1/1,000 |
| Anti-Mouse HRP | Jackson ImmunoResearch (115-035-166) | Goat (polyclonal) | 1/5,000 |
| Anti-Rabbit HRP | Jackson ImmunoResearch (115-035-044) | Goat (polyclonal) | 1/5,000 |
| Anti-Mouse Alexa Fluor 488 | Invitrogen (A-11029) | Goat (polyclonal) | 1/1,000 |
| Anti-Mouse Alexa Fluor 594 | Invitrogen (A-11032) | Goat (polyclonal) | 1/1,000 |
| Anti-Mouse Cy5 | Jackson ImmunoResearch (115-175-146) | Goat (polyclonal) | 1/500 |
| Anti-Rabbit Alexa Fluor 488 | Invitrogen (A-11034) | Goat (polyclonal) | 1/1,000 |
| Anti-Rabbit Alexa Fluor 594 | Invitrogen (A-11037) | Goat (polyclonal) | 1/1,000 |
| Anti-Rabbit Cy5 | Jackson ImmunoResearch (111-175-144) | Goat (polyclonal) | 1/1,000 |

experiments were approved by the Saarland state's "Landesamt für Gesundheit und Verbraucherschutz" animal license number 36/2016.

## Golgi staining

The 2-month-old anesthetized mice were dislocated prior to brain dissection, and 100 μm thick coronal brain sections were cut on a vibratome in the hippocampal region. The dendritic spines of hippocampal neurons from the CA1 *stratum radiatum* were visualized by the Golgi impregnation technique. For this, we used the FD Rapid GolgiStain kit (PK401, FD Neuro-Technologies). Brain sections were immersed in equal volumes of solutions A and B for 7 d and impregnated with solution C for 48 h at 4°C. Then, the sections were washed twice in double-distilled water and incubated for 10 min in a mixture solution D and solution E in double-distilled water (1:1:2). Sections were washed twice, dehydrated with increasing concentrations of ethanol, and mounted with epoxy resin (Fluka). Stacks of bright-field images with 0.3 μm spacing were acquired with a Zeiss Axioskop 50 microscope with 63× oil objective (NA 1.4; Plan-Apochromat) coupled to a CCD camera (CoolSnap ES; Roper Scientific) operated by Metaview software (Molecular Devices). Images were analyzed with ImageJ. The number of dendritic spines (>1 μm protrusion) along portion of dendrite of 100 μm was counted with ImageJ.

## Transmission electron microscopy of the CA1 hippocampus

The 2-month-old mice were anesthetized and intracardially perfused with phosphate-buffered 0.9% NaCl (PBS), followed by 0.1 M phosphate buffered 4% paraformaldehyde (pH 7.4), supplemented with 0.05% glutaraldehyde (G6257, Sigma-Aldrich). The brains were carefully removed, postfixed for 4 h in the same fixative, and 60 μm sections were cut with a vibratome. After several washes in PBS, the sections were postfixed in 1% glutaraldehyde in the same buffer for 10 min and processed for EM. This included treatment with osmium tetroxide (1%

in 0.1 M PB), block staining with uranyl acetate, dehydration through a graded series of ethanol, and flat embedding on glass slides in Durcupan (Fluka) resin. Regions of interest were cut at 70 nm on an ultramicrotome (Reichert Ultracut E; Leica) and collected on 1-slot copper grids. Staining was performed on drops of 1% aqueous uranyl acetate, followed by Reynolds's lead citrate. EM images were acquired in a JEOL-1200 electron microscope with a digital camera (Veleta, SIS; Olympus) and analyzed with ImageJ. Twenty images per animal from 3 animals per genotype were used for quantification. Measurements were performed on 2D section through the synaptic bouton/spine. The number of synapses per $\mu m^2$ was calculated. A synapse was considered if it met 3 criteria: a presynaptic bouton filled with at least 10 SVs (1) juxtaposed to the head of a dendritic spine with a clearly visible PSD (2) and the presence of the neck in the section (3). Number of SVs and areas of presynaptic boutons were quantified in each synapse using the free-shape tool and the cell counter plugins of ImageJ. We used the straight tool of ImageJ to measure the lengths of PSDs, and head and neck diameters. Note that the head diameter was taken parallel to the PSD, and the neck diameter was perpendicular to the neck membranes.

## Electrophysiological recordings in CA1

**Field recordings.** Alix wt and ko brain slices were prepared from 2-month-old C57BL/6 wt and Alix ko mice. The brains were removed quickly and 350-μm thick sagittal slices containing both cortex and hippocampus were cut in ice-cold sucrose solution (2.5 mM KCl, 1.25 mM $NaH_2PO_4$, 10 mM $MgSO_4$, 0.5 mM $CaCl_2$, 26 mM NaHCO3, 234 mM sucrose, 11 mM glucose, and saturated with 95% $O_2$ and 5% $CO_2$) with a Leica VT1200 blade microtome (Leica Microsystemes, Nanterre, France). After cutting, hippocampi were extracted from the slice and transferred to oxygenated artificial cerebrospinal fluid (ACSF: 119 mM NaCl, 2.5 mM KCl, 1.25 mM $NaH_2PO_4$, 1.3 mM $MgSO_4$, 2.5 mM $CaCl_2$, 26 mM $NaHCO_3$, 11 mM glucose) at $37 \pm 1°C$ for 30 min and then kept at room temperature (RT) for at least 1 h before recordings. Each slice was individually transferred to a submersion-type recording chamber and continuously superfused (2 ml/min) with oxygenated ACSF. Extracellular recordings were obtained at 28°C from the apical dendritic layers of the hippocampal CA1 area, using glass micropipettes filled with ACSF. Field excitatory postsynaptic potentials (fEPSPs) were evoked by the electrical stimulation of Schaffer collaterals afferent to CA1. The magnitude of the fEPSPs was determined by measuring their slope. Signals were acquired using a double EPC 10 Amplifier (HEKA Elektronik Dr. Schulze GmbH, Germany) and analyzed with Patchmaster software (HEKA Elektronik Dr. Schulze GmbH, Germany). For the induction of LTP, test stimuli were delivered once every 15 s. Stimulus intensities were adjusted to produce 40% to 50% of the maximal response. A stable baseline was recorded for at least 15 min. LTP was induced by high frequency stimulation (4 trains delivered at 100 Hz with 5 min between each train). Average value of fEPSP slope was expressed as a percentage of the baseline response.

**Patch clamp recordings.** Alix[fl/fl] mice [8] with heterozygous cre expression (Emx1[IREScre] (Emx[tm1(cre)Krj], MGI: 2684610)) [37] as well as control mice (Alix[fl/fl] x Emx[wt]) were decapitated under anesthesia, and brains were removed from the skull and immersed in an ice-cold, oxygenated (5% $CO_2$/95% $O_2$ (pH 7.4)) slice preparation solution containing (in mM) 87 NaCl, 3 KCl, 25 $NaHCO_3$, 1.25 $NaH_2PO_4$, 3 $MgCl_2$, 0.5 $CaCl_2$, 75 sucrose, and 25 glucose. Coronal slices of 300 μm thickness were prepared with a vibratome (Leica VT 1200S, Nussloch, Germany) and transferred to a nylon basket slice holder for incubation in ACSF containing (in mM) 126 NaCl, 3 KCl, 25 $NaHCO_3$, 15 glucose, 1.2 $NaH_2PO_4$, 2 $CaCl_2$, and 2 $MgCl_2$ at 32°C for 30 min. Subsequently, slices were removed from the water bath and kept at RT with continuous oxygenation prior to use.

Slices were transferred to the recording chamber continuously perfused with oxygenated ACSF containing (in mM) 1 $MgCl_2$ and 2.5 $CaCl_2$ (2 to 5 ml/min). During recordings, 50 μM strychnine and 50 μM picrotoxin were added to block inhibitory synaptic transmission. Pyramidal neurons were identified morphologically using the recording microscope (Axioskop 2 FS mot, Zeiss, Jena, Germany) with a 40× water immersion objective. Images were detected with a QuantEM 512SC camera (Photometrics, Tucson, United States of America). Whole-cell membrane currents were recorded by an EPC 10 USB amplifier (HEKA, Lambrecht, Germany), low pass filtered at 3 kHz and data acquisition was controlled by Patchmaster software (HEKA). The resistance of patch pipettes (7 to 9 ΩM) were fabricated from borosilicate capillaries (OD: 1.5 mm; Sutter, USA) using a Micropipette Puller (Model P-97, Sutter Instrument, California, USA). Patch pipettes were filled with a solution containing (in mM) 125 cesium gluconate, 20 tetraethylammonium (TEA), 2 $MgCl_2$, 0.5 $CaCl_2$, 1 EGTA, 10 HEPES, and 5 $Na_2ATP$ (pH 7.2). The holding potential in voltage-clamp mode was at −70 mV. sEPSCs were recorded under holding potential for 40 s in 5 min later after whole-cell configuration. Here, only < = −5 pA sEPSC was analyzed. EPSCs of pyramidal neurons in the hippocampal CA1 were evoked by stimulating Schaffer collaterals of CA3 neurons with a concentric bipolar microelectrode (MicroProbes, USA). Stimulus duration was 200 μs. To estimate short-term plasticity, 25 trains of stimuli were applied to induce synaptic depression, and the depressing trains were repeated at 10 Hz with an interval of 20 s to allow for recovery from synaptic depression. To investigate recovery from depression, stimulation with gradually increased interval was applied. The stimulation threshold was applied from 30 μA to 80 μA. All the experiments were conducted at RT (22 to 24˚C).

Data generated by Patch Master were loaded into Matlab (Mathworks, Massachusetts, USA) with a module adapted from sigTOOL [55]. EPSC traces from the same cells were manually checked and pooled. The average of EPSC traces from each cell was used to analysis. The RRP size was estimated by using train-extrapolation method [56,57].The cumulative sum of the peak EPSC amplitudes was plotted against the stimulation numbers. The last 10 data points were linearly fitted. The RRP size was calculated by back-extrapolating the fitted line to the y-axis (stimulation ID = 0). The release probability was calculated by $p = \text{Amplitude}^1/\text{RRP}_{train}$. The fitted slop represents the replenishment rate [58]. The recovery was estimated with the decay constant (τ) as $N_t = N_0 * exp\left(-\frac{t}{\tau}\right)$.

Data analysis was performed using routines that were custom written in Matlab.

## EEG telemetry and unilateral intracortical kainate injection

We took advantage of the unilateral intracortical kainate injection model for human temporal lobe epilepsy. Telemetric EEG transmitter implantation, kainate injection, and data analysis was adapted from Bedner and colleagues [39].

Controls and alix cko were implanted with telemetric EEG transmitters (DSI PhysioTel ETA-F10, Harvard Biosciences, Holliston, Massachusetts, USA). The animals were placed in a stereotaxic frame (Robot stereotaxic, Neurostar, Tübingen, Germany) for implantation of depth electrodes at 3.4 mm posterior to bregma and bilaterally 1.6 mm from the sagittal suture. After postsurgical care and recovery, mice were again placed in the stereotaxic frame and injected with 70 nl of a 20 mM solution of kainate (Tocris, Wiesbaden-Nordenstadt, Germany) in 0.9% NaCl, above the right dorsal hippocampus (1.9 mm posterior to bregma, 1.5 mm from sagittal suture, and 1.3 mm from skull surface). Kainate was injected at a rate of 70 nl/min with a 10 μl Nanofil syringe (34 GA blunt needle, World Precision Instruments, Sarasota, Florida, USA). The syringe was kept in place for 2 min after the injection was completed to avoid liquid reflux.

Cages were placed on individual radio receiving plates (DSI PhysioTel RPC-1, Data Sciences International, St. Paul, USA), which record EEG signals and sent them, together with the video recording (MediaRecorder Software, Noldus Information Technology, Wageningen, the Netherlands), to an input exchange matrix (DSI PhysioTel Matrix 2.0 (MX2), Ponemah software, DSI, Data Sciences International, St. Paul, USA). The animals were monitored for at least 20-h post kainate injection. In our model, the mortality rate associated to SE is less than 5% in more than 50 mice with different genetic backgrounds over the last 12 months.

EEG traces were analyzed with the Neuroscore software (Version 3.3.1., Data Sciences International, St. Paul, USA). Electrographic seizures were detected with the spike detection protocol. Subsequently, an additional manual screen was employed to remove artifacts that were eventually picked up. Seizures were characterized by high frequency spiking and ceased with a postictal depression (flattening of EEG). Seizure detection was complemented by synchronized video monitoring. Electrographic seizures were associated with behavioral analogs of Racine stages II to V [59]. The total duration of SE was defined from the first electrographic seizure to the first seizure free period lasting 1 h.

For immunohistochemistry analysis, 40-μm free-floating vibratome sections were incubated for 1 h in blocking buffer (5% HS, 0.3% Triton X in 1× PBS) at RT. Sections were incubated with primary antibodies (see Antibodies section) diluted in blocking solution, overnight at 4°C. Secondary antibodies and DAPI (0.025 μg/ml, D9542, Sigma-Aldrich) were diluted in blocking buffer and incubated for 2 h at RT.

## Cell culture and transfection

Cortical and hippocampal neurons from rat E18 embryos [60], P0 Alix wt/ko mice [8], and P0 controls/Alix cko mice [61] were prepared as previously described. Briefly, cortices and hippocampi were dissected from E18 rat embryos or P0 mouse pups, treated with trypsin or papain, and mechanically dissociated. Dissociated cells were seeded at a density of $5 \times 10^4/cm^2$ in 100 mm dishes for cortical neurons and $1.5 \times 10^4/cm^2$ onto acid-washed coverslips (either 14 mm diameter or 25 mm; Marienfeld) in either 4-well plates or P35 dishes (Thermo Fisher Scientific) precoated for 4 h with 50 μg/ml poly-D-lysine (P6407, Sigma-Aldrich). Neurons were plated in DMEM containing 10 unit/ml penicillin, 10 μg/ml streptomycin, 0.5 mM L-glutamine, and 10% of inactivated horse serum (Invitrogen) Neurons were maintained in water-saturated 95% air/5% $CO_2$ at 37°C. The seeding medium was replaced after 20 h with serum-free neuronal culture medium (Neurobasal medium containing 2% B27 supplement, 10 unit/ml penicillin, 10 μg/ml streptomycin, and 0.5 mM L-glutamine; Invitrogen). Neurons were transfected at 10 day in vitro (DIV) as previously described [62]. Briefly, for each P35 dish, 2 μg plasmid DNA, 250 mM $CaCl_2$ were mixed with an equal volume of 2× BES-buffered saline and left to precipitate for 20 min at RT. Neurons were placed in transfection medium (Minimum Essential Medium containing 0.22% $NaHCO_3$, 20 mM D-glucose, and 0.5 mM L-glutamine) supplemented with 2% B27, before the DNA precipitate was added. They were then incubated for 1.5 h at 37°C and 5% $CO_2$. Neurons were then washed by being placed in transfection medium (pre-warmed to 37°C in 10% $CO_2$) for 20 min at 37°C and 5% $CO_2$. Finally, they were transferred back into their conditioned medium.

Primary cultures of CGNs were prepared from 6-day-old C57BL/6 Alix wt and Alix ko pups as described previously [27], with some modifications. The cerebella were removed, cleared of their meninges, and cut into 1-mm pieces. They were then incubated at 37°C for 10 min in 0.25% trypsin-EDTA and DNAse (1,500 U/ml). Trypsin was inactivated, and cells were dissociated in culture medium (DMEM containing 10% fetal bovine serum, 2 mM L-glutamine, 25 mM KCl, 10 mM HEPES, 10 unit/ml penicillin, and 10 μg/ml streptomycin).

After filtration on 70-μm cell strainers, neurons were plated at $5.10^5$ cell/$cm^2$ onto poly-D-lysine (10 μg/ml, P6407, Sigma-Aldrich) precoated coverslips. Cytosine-β-D-arabinoside (10 μM, C1768, Sigma-Aldrich) was added after 1 DIV, to prevent the growth of non-neuronal cells, until 8 DIV when neurons were used for activity dependent bulk endocytosis experiment (see below).

## Multielectrode array

Dissociated hippocampal neurons were resuspended in neuronal medium and plated at a $10^6$ cells/$cm^2$ on poly-L-lysine-coated MEAs comprising 59 extracellular recording electrodes and 1 reference electrode (MEA-60 Multichannel Systems, MCS, Germany). Electrodes were 30 μm in diameter and separated by a distance of 200 μm. MEA-60 plates were connected to a 60-channel data acquisition system (USB-MEA64) and associated amplifier (MEA1060-Inv-BC) powered by a PS40W power supply system (MCS). The recording system was then placed in a humidified incubation chamber at 37°C and 5% $CO_2$. Neurons were left for 5 min to equilibrate before recording. Basal spontaneous activity was recorded for 3 min prior addition of drugs. Bicuculline (100 μM) and 4-aminopyridine (5 mM) were then added and neuronal activity recorded for another 10 min. Signals were recorded with a 1,100 gain, sampled at 20 kHz, and analyzed with MC Rack software (MCS). Raw signals were first filtered with a Butterworth band pass filter (second order) between 200 and 800 Hz to remove electrical noise and low frequency signals. Spike detection analysis was performed on filtered signals using a threshold of 6 standard deviations to mark out action potentials. Burst events were identified as a minimum of 5 consecutive spikes with an inter-spike interval lower than 100 ms. The minimum interval time between 2 bursts was fixed at 1,000 ms. Representative traces were exported using custom-made Matlab functions (Matlab 2014b).

## Live imaging of synaptophysin-pHluorin upon electrical stimulation

Hippocampal neurons from Alix wt and ko mice were transfected with Syp-pH at 6 DIV. The recycling of SVs was imaged in a buffer solution containing 120 mM NaCl, 5 mM KCl, 2 mM $CaCl_2$, 2 mM $MgCl_2$, 5 mM glucose, 10 mM HEPES adjusted to pH 7.4, and 270 mOsm/l. Experiments were carried out at 34°C. Neurons were stimulated by electric field stimulation (platinum electrodes, 10 mm spacing, 1 ms pulses of 50 mA, and alternating polarity at 5 to 40 Hz) applied by a constant current stimulus isolator (SIU-102, Warner Instruments). The presence of 10 μM 6-cyano-7-nitroquinoxaline-2,3-dione (CNQX) and 50 μM D,L-2-amino-5-phosphonovaleric acid (AP5) prevented recurrent activity. Experiments were performed on an inverted microscope (IX83, Olympus) equipped with an Apochromat N oil 100× objective (NA 1.49). Images were acquired with an electron multiplying charge coupled device camera (QuantEM:512SC; Roper Scientific) controlled by MetaVue7.1 (VS3200, Roper Scientific). Samples were illuminated by a 473-nm laser (Cobolt). Emitted fluorescence was detected after passing a 525/50-nm filter (Chroma Technology). Time-lapse images were acquired at 1 or 2 Hz with integration times from 50 to 100 ms.

Image analysis was performed with custom macros in Igor Pro (Wavemetrics) using an automated detection algorithm as described previously [63]. The image from the time series showing maximum response during stimulation was subjected to an "à trous" wavelet transformation. All identified masks and calculated time courses were visually inspected for correspondence to individual functional boutons. The intensity values were normalized to the ten frames before stimulation.

## Live fluorescence imaging of Alix and endophilin recruitment to synapses

For live imaging of protein recruitment, cultured hippocampal neurons were co-transfected with Alix-mCherry or endophilin-A2-mCherry together with Syp-pH expression vectors 2 to 4 days prior to imaging. All live imaging experiments were performed at 37°C, and images were acquired using a spinning disk confocal microscope (AxioObserver Z1) with a 63× oil objective (NA 1.46, Zeiss). Transfected neurons were placed in basal medium for 10 min and then mounted in an imaging chamber (POC-R2 Cell cultivation system, Zeiss). After 2 min in basal medium, a 5× Bic/4AP solution was added (final concentration, 100 μM and 5 mM, respectively) for stimulation which lasted 5 min, and the chamber was perfused with basal medium at 3 ml/min for washing for 5 min. Region of interest (ROI) were drawn on "presynapses" defined by spots of Syp-pH that increased during stimulation. Fluorescence values were measured in these ROI and then normalized to the initial fluorescence values (fluorescence values prior to stimulation) using ImageJ.

**Quantification of Alix recruitment during live imaging.** Synapses were identified as Alix-positive before stimulation. The intensity of each individual synapses was measured (ROI $20 \times 20$ pixel) at each time point recorded during the live imaging. Synapses were counted as responding to stimulation when Syp-pH increase was bigger than $4 \times$ SD of the non-stimulated synapses. Similarly, Alix accumulation was considered as significant when Alix fluorescence intensity increased over $4 \times$ SD of the non-stimulated synapses.

## Immunofluorescence

Cultured hippocampal neurons were fixed for 20 min at RT in phosphate-buffered 4% paraformaldehyde supplemented with 4% sucrose. After 3 washes in PBS, cells were permeabilized and blocked in PBS containing 0.3% Triton X-100 and 3% BSA for 15 min at RT. Coverslips were incubated for 1 to 2 h at RT with primary antibodies (see "Antibodies" section) diluted in the blocking solution. After washing in PBS, cells were incubated for 1 h with secondary antibodies conjugated to Alexa Fluor 488, Alexa Fluor 594, or Cyanine 5 (Cy5), diluted in the blocking solution. Coverslips were rinsed and mounted in Mowiol. Images were acquired on a Leica SPE microscope using a 40× dry objective (NA 0.75, Leica) or a 100× oil immersion objective (NA 1.4, Leica) at 488 nm, 532 nm, or 635 nm.

**Synapse density.** Cultured hippocampal neurons were fixed and stained with antibodies at 14 to 16 DIV as described above. Presynaptic boutons (synapsin-1-positive) and postsynaptic terminals (PSD95-positive) were selected using the Spot Detector plugin in Icy software [64] (wavelet detection with size filtering between 0.4 μm and 2 μm in diameter) on max image projections. Synapses were defined as spots of colocalization between the detected presynaptic and postsynaptic terminals that were within 3 μm of each other. Synapses were counted using the Colocalization Studio in ICY software [65].

**Colocalization.** Colocalization between Alix and PSD95 or Alix and synapsin1 was assessed by using the "coloc 2" plugin of Fiji on fixed ROI ($70 \times 15$ μm). The Pearson's correlation coefficient was used to define the colocalization between Alix and either PSD95 or synapsin1.

**Distance between Alix and syn/PSD95.** The distances between Alix and synapsin1 or Alix and PSD95 were measured at the level of individual synapses by using the Fiji line scan and plot profile tools to measure the distance between the 2 peaks of intensity. Each distance between intensity peak of Alix versus synapsin and Alix versus PSD95 was calculated and represented as dot plots.

**Protein recruitment assays.** Cultured hippocampal neurons were transfected with expression vectors 24 to 48 h prior to stimulation. The transfected neurons were incubated in

basal medium (150 mM NaCl, 5 mM KCl, 1.3 mM $CaCl_2$, 10 mM HEPES, and 33 mM D-glucose at pH 7.4) for 10 min prior to stimulation. Coverslips were either treated with basal medium alone or Bic/4AP solution (basal medium supplemented with 100 μM bicuculline and 5 mM 4-aminopyridine) for 5 min at 37˚C and 5% $CO_2$. Coverslips were fixed and stained with antibodies against synapsin-1 and PSD95 (see "Antibodies" section). Images were acquired on a Leica SPE microscope using a 100× oil immersion objective (NA 1.4, Leica). The acquired images were analyzed using ImageJ. ROI were drawn on areas of axons that colocalize with anti-synapsin-1 spots, but not with anti-PSD95 spots to ensure presynaptic measurement. Fluorescence intensities were then measured in these ROI and normalized to fluorescence intensities measured on other regions of axons to give the relative fluorescence intensity at presynaptic boutons.

## Calcium increase imaging upon neuronal stimulation

Hippocampal neurons cultured on 24-well plates were placed in basal medium and incubated with 1 μM Fluo-4-AM for 1 h at 37˚C. After washing, neurons were left in 250 μl basal medium for 5 min. Plates were then transferred to a Pherastar automatic plate reader (BMG Labtech, Germany) set to 37˚C, to record fluorescence intensity at 0.1 Hz during 10 min, for each well, and with fluorescence measurement settings as described on http://www.bmglabtech.com/media/35216/1043854.pdf. Neurons were stimulated with 50 μl 6x Bic/4AP solution (final concentration, 100 μM and 5 mM, respectively) after 5 min with automatic injection.

## Dextran uptake in hippocampal synapses upon neuronal stimulation

The protocol for dextran uptake was adapted from [66]. A total of 15 to 17 DIV hippocampal neurons were washed briefly in extracellular solution (ECS [mM] 150 NaCl, 5 KCl, 10 mM HEPES, 33 mM D-Glucose, 1.3 $CaCl_2$, 1 $MgCl_2$ (pH 7.4)) and stimulated with Bic/4AP solution for 5 min or with K50 solution (K50 [mM]: 110 NaCl, 50 KCl, 1.3 $CaCl_2$, 1 $MgCL_2$, 10 HEPES, 33 D-Glucose (pH 7.4)) for 90 s, at 37˚C, and 5% $CO_2$, in the presence of 10 kDa tetramethylrhodamine-dextran (50 μM). To remove excess dextran, neurons were subsequently washed 4 times in washing solution (ECS supplemented with 0.2% BSA and warmed to 37˚C). For ADBE inhibition, neurons were stimulated in the presence of 2 μM GSK3 inhibitor (CT99021, Tocris) and placed in fresh basal medium containing 2 μM GSK3 inhibitor for 10 min at 37˚C and 5% $CO_2$. Neurons were then fixed as previously described and imaged with a Leica SPE microscope using a 40× dry immersion objective (NA 0.75, Leica) or a Zeiss LSM 880 using a 63× oil objective (NA 1.4, Plan Apochromat). Analysis was performed on ImageJ. The number of fluorescent spots was counted in a defined field of view (130 μm × 130 μm) in thresholding analysis with a diameter limit between 300 nm and 2 μm (resolution limit for the microscope and maximum size of a nerve terminal).

For rescue experiments, cultured hippocampal neurons were transfected 2 to 4 d prior to the day of experiment with the following constructs: Alix wt, Alix I212D (AlixΔCHMP4B), AlixΔPGY (AlixΔALG-2), and Alix R757E (AlixΔendo). Dextran uptake assay was performed as described above. Images were acquired on a Leica SPE microscope using a 40× oil immersion objective (NA 1.25, Leica) at 488 nm and 532 mm excitation. The analysis was performed on ICY software. ROIs were generated closely around axons by using "thresholder." Then dextran spots within these ROI were counted by using "spot detector" with size limit between 300 nm and 2 μm. The length of axon per field of view was estimated by manually drawing "Polyline type ROI" over the axon images. The number of dextran spots per μm of axon was calculated and expressed as ratio to control values.

## EM examination of ADBE in culture neurons

Analysis of ADBE from cerebellar granule neurons was performed as described previously with some modifications [32]. Eight DIV CGN were incubated in hyperpolarizing medium (170 mM NaCl, 3.5 mM KCl, 0.4 mM $KH_2PO_4$, 20 mM TES, 5 mM $NaHCO_3$, 5 mM D-glucose, 1.2 mM $Na_2SO_4$, 1.2 mM $MgCl_2$, 1.3 mM $CaCl_2$, pH7.4) for 10 min prior to stimulation. Neurons were then incubated for 2 min with 10 mg/ml HRP (Sigma P8250) in either the hyperpolarizing medium or a high-potassic solution containing 50 mM KCl (123.5 mM NaCl, 50 mM KCl, 0.4 mM $KH_2PO_4$, 20 mM TES, 5 mM $NaHCO_3$, 5 mM D-glucose, 1.2 mM $Na_2SO_4$, 1.2 mM $MgCl_2$, 1.3 mM $CaCl_2$, pH7.4) before rapid washing in PBS and fixation in PBS-glutaraldehyde 2% for 30 min. After 3 washes in Tris buffer 100 mM, endocytosed HRP was revealed by incubation in Tris 100 mM containing 0.1% diaminobenzidine and 0.2% $H_2O_2$. The cultures were then post-fixed in 1% osmium tetroxide, dehydrated, and embedded in Epon. Synapses were photographed with a JEOL-1200 electron microscope. Quantification was performed as follows: HRP-positive structures were quantified per synapse. Their diameters were measured manually using the straight-line tool of imageJ. Based on these measurements, HRP-positives structures were classified as SVs when the diameter was less than 100 nm and as bulk endosome when the diameter was more than 100 nm.

For FIB-SEM, the block was mounted on a pin, coated with gold, and inserted into the chamber of a HELIOS 660 Nanolab DualBeam SEM/FIB microscope (FEI, Eindhoven, the Netherlands). ROI was prepared using focused ion beam (FIB) and ROI set to be approximately 15 μm wide. During the acquisition process, the image acquisition parameters of the electron beam were 2 kV and 0.4 nA and the thickness of the FIB slice between each image acquisition was 10 nm. The segmentation of the synapse and ADBE was done with Amira software (2020.3, Thermo Fisher Scientific) and the movie obtained using Imaris (Oxford Instruments).

## Synaptosomal preparation from cortical neurons

Synaptosome-enriched membranes from 15 DIV cortical neurons were prepared as described previously with some modifications [67]. Briefly, cultured neurons were stimulated for 15 min with a mixture of Bic/4AP (50 μM/2.5 mM). After a wash in HBSS, neurons were homogenized by passing 15 to 20 times through 0.25G needle in cold buffer containing 0.32 M sucrose, 10 mM HEPES, 15 mM NaF, 15 mM β-glycerophosphate and protease inhibitors (Roche) (pH 7.4). Samples were maintained at 4°C during all steps of the experiment. Homogenates were cleared at 1,000 g for 10 min to remove nuclei and large debris. The resulting supernatants were spun down at 12,000 g for 20 min to obtain a crude membrane fraction and washed twice in HEPES buffer 4 mM containing 1 mM EDTA, 15 mM NaF, 15 mM β-glycerophosphate and protease inhibitors (Roche) (pH 7.4). The resulting pellet was solubilized in 0.5% Triton X-100, 20 mM HEPES, 100 mM NaCl, 15mM NaF, 15 mM β-glycerophosphate (pH 7.2), containing protease inhibitors (Roche) for 20 min at 4°C with mild agitation and analyzed by western blot.

## Western blot

Cells lysates were resuspended in Laemmli buffer and resolved by SDS-PAGE in 10% polyacrylamide gels. Proteins were electro-transferred onto PVDF membranes that were then blocked for 30 min in TBS containing 0.1% Tween 20 and 5% dry milk and incubated for 1 h to overnight with primary antibodies diluted in the blocking solution. After washes in TBS–Tween, the membranes were further incubated for 1 h with secondary antibodies coupled to HRP, washed as before, and incubated with luminescence-generating HRP substrate. Bound antibodies were revealed by luminography on film.

## Statistical analysis

The comparison of 2 groups was performed using 2-sided Student *t* test or its non-parametric correspondent, the Mann–Whitney test, if normality was rejected (Shapiro–Wilks test). The comparisons of more than 2 groups were made using 1- or 2-way ANOVAs followed by post hoc tests (Holm Sidak's or Tukey's HSD) to identify all the significant group differences. *N* indicates independent biological replicates. The graphs with error bars indicate 1 SEM (+/−) except for Supporting information S5 Fig, where we used box plots showing distribution of medians (whiskers = min and max values), Figs 5 and S7 where median and interquartile range (IQR) were represented. The significance level is denoted as usual ($^*p < 0.05$, $^{**}p < 0.01$, $^{***}p < 0.001$). All the statistical analyses were performed using Prism7 (Graphpad version 7.0a, April 2, 2016).

## Supporting information

**S1 Fig.** (**A**) No difference in calcium rise in Alix wt and ko hippocampal neurons upon Bic/4AP stimulation. Fluo4-AM intensity corresponds to the change in fluorescence, normalized to initial fluorescence (*N* = 4 experiments, *p* = 0.9999, 2-way ANOVA). (**B**, **C**) Alix-YFP does not accumulate in dendritic spines upon Bic/4AP stimulation of 15 DIV hippocampal neurons. Scale bar: 5 μm. Postsynaptic Alix-YFP corresponds to the ratio between YFP-fluorescence at PSD95 labeled ROI and at neighboring dendritic parts. Scale bar: 5 μm. Average +/− SEM, *N*, statistical analysis: (**A**) *N* = 4 experiments, *p* = 0.9999, 2-way ANOVA. (**C**) 1.34 +/− 0.24; 1.25 +/− 0.09 for no stim and stim, respectively. *N* = 33 and *N* = 25 neurons for both conditions, from 5 experiments, *p* = 0.3821, Mann–Whitney test. The data underlying all the graphs shown in the figure can be found in S1 Data. Alix, ALG-2-interacting protein X; DIV, day *in vitro*; ko, knockout; wt, wild type.
(PDF)

**S2 Fig. Calcium chelators (BAPTA-AM and EGTA-AM) block presynaptic Alix recruitment upon Bic/4AP stimulation.** Presynaptic Alix-YFP corresponds to the ratio of fluorescence between presynaptic and non-synaptic axonal ROI. Average +/− SEM, *N*, statistical analysis: 1.63 +/− 0.12; 3.27 +/− 0.44; 1.58 +/− 0.10; 1.54 +/− 0.009; 1.69 +/− 0.13; 1.43 +/− 0.04 for DMSO no stim, DMSO stim, EGTA no stim, EGTA stim, BAPTA no stim, BAPTA stim respectively. *N* = 12, 9, and 9 neurons for DMSO, EGTA, and BAPTA, respectively. *p* = 0.0001, 1-way ANOVA. The data underlying all the graphs shown in the figure can be found in S1 Data. Alix, ALG-2-interacting protein X; ROI, Regions of interest.
(PDF)

**S3 Fig.** (**A, B**) Quantification of the syp-pH peak of fluorescence (PeakDF) during stimulation at 5 Hz (A) and 40 Hz (B) normalized to the F0 showing no significant difference in syp-pH exocytosis between Alix wt and Alix ko both at 5 Hz and 40 Hz. (**C–E**) Full dot plot representation of SV (blue dots C, E) and bulk endosome (red dots, C, D) diameters in Alix wt and Alix ko neurons in basal or stimulated condition showing that bulk endosomes of Alix ko synapses are smaller than in Alix wt neurons, related to Fig 3K. Average +/− SEM, *N*, statistical analysis: (**A**) 0.6141 +/− 0.078, 0.5184 +/− 0.044 for Alix wt and Alix ko, respectively. *N* = 20 and 38 for Alix wt and Alix ko, respectively, *p* = 0.4028, Mann–Whitney test. (**B**) 2.030 +/− 0.2281, 1.84 +/− 0.162 for Alix wt and Alix ko, respectively. *N* = 24 and 45 for Alix wt and Alix ko, respectively, *p* = 0.4854, Unpaired *t* test. (C) 112.2 +/− 2.37 nm; 191.7 +/− 3.39 nm; 152.4 +/− 4.1 nm; 135.2 +/− 1.79 nm for Alix wt no stim, Alix wt stim, Alix ko no stim, Alix ko stim, respectively. *N* = 304, 1,326, 285, 2,415 vesicle for Alix wt no stim, Alix wt stim, Alix ko no stim, Alix ko stim, respectively, from 3 independent experiments. *p* < 0.0001 in all conditions tested,

Kruskal–Wallis test. (**D**) 133.8 +/− 2.99 nm; 223.3 +/− 3.82 nm; 171.5 +/− 4.6 nm; 179.9 +/− 2.60 nm for Alix wt no stim, Alix wt stim, Alix ko no stim, Alix ko stim, respectively. $N$ = 181, 1,031, 223, 1,354 bulk endosomes for Alix wt no stim, Alix wt stim, Alix ko no stim, Alix ko stim, respectively, from 3 independent experiments. $p < 0.0001$ in all conditions tested, Kruskal–Wallis test. (**E**) 81.7 +/− 1.61 nm; 82.2 +/− 0.76 nm; 85.4 +/− 1.3 nm; 78.8 +/− 0.39 nm for Alix wt no stim, Alix wt stim, Alix ko no stim, Alix ko stim, respectively. $N$ = 131, 309, 69, 1,090 SVs for Alix wt no stim, Alix wt stim, Alix ko no stim, Alix ko stim, respectively, from 3 independent experiments. The data underlying all the graphs shown in the figure can be found in S1 Data. Alix, ALG-2-interacting protein X; ko, knockout; SV, synaptic vesicle; syp-pH, synaptophysin-pHluorin; wt, wild type.
(PDF)

**S4 Fig.** (**A**) Confocal images of control and Alix cko hippocampal neurons unstimulated (left) and stimulated (right + inset) in the presence of 10 kDa dextran (red). Scale bars: 50 μm and 10 μm (inset). (**B**) Quantification of the dextran uptake showing a significant reduction of uptake in cko neurons upon stimulation with 50 mM KCl for 90 s. (**C, D**) Dextran uptake is abolished in Alix wt neurons treated with an inhibitor of bulk endocytosis (GSK3-inhibitor). Confocal images of Alix wt hippocampal neurons stimulated in the presence of 10 kDa dextran with or without a GSK3 inhibitor. Scale bar: 50 μm. The % dextran uptake corresponds to the number of dextran spots per ROI expressed as percentages of the positive control. (**E, F**) Multiple electrode array activity recordings of 15 DIV hippocampal neuron cultures showing the effect of Bic/4AP incubation for 10 min. Representative traces of wt (black) and ko (gray) cultures are shown on panel E. (**G**) Dextran uptake is abolished in wt neurons by calcium chelators EGTA and BAPTA. The % dextran uptake corresponds to the number of dextran spots per ROI expressed as percentages of the positive control for each experiment. Average +/− SEM, $N$, statistical analysis: (**B**) 5.80 +/− 0.50; 5.19 +/− 0.39; 23.54 +/− 1.361; 7.529 +/− 1.476 for controls no stim, cko no stim, controls stim, cko stim, respectively. $N$ = 27, 23, 30, 21 controls no stim, cko no stim, controls stim, cko stim, respectively, from 4 independent experiments, $p < 0.0001$, 1-way ANOVA. (**C**) 12.41 +/− 1.51; 100 +/− 11.29; 12.46 +/− 1.90; 12.40 +/− 2.99 for DMSO no stim, DMSO stim, GSK3-inhibitor no stim, GSK3-inhibitor stim, respectively. $N$ = 3 experiments, $p = 0.0001$, 1-way ANOVA. (**F**) average +/− SEM are as follow: 8.81 +/− 1.05; 18.48 +/− 1.92; 2.41 +/− 0.27; 10.33 +/− 1.23 for Alix wt no stim, Alix wt stim, Alix ko no stim, Alix ko stim, respectively. $N$ = 53 and 58 field of view for Alix wt and Alix ko, respectively, $p < 0.0001$, 1-way ANOVA. (**F**) 100 +/− 0; 21.89 +/− 4.19; 18.34 +/− 3.54; 14.53 +/− 4.17 for Alix wt DMSO, Alix wt EGTA, Alix wt BAPTA, and Alix ko DMSO, respectively. $N$ = 3 experiments, $p = 0.0001$, 1-way ANOVA. The data underlying all the graphs shown in the figure can be found in S1 Data. Alix, ALG-2-interacting protein X; cko, conditional ko; DIV, day *in vitro*; ROI, Regions of interest; wt, wild type.
(PDF)

**S5 Fig. Synaptic contacts are unaffected in Alix neurons.** (**A**) Representative electron micrographs of the CA1 *stratum radiatum* from Alix wt and ko brain sections. Presynaptic profiles are highlighted in blue, and dendritic spines are in purple. Scale bar: 2 μm. Graph shows no difference in synaptic density per μm$^2$ of Alix wt and ko brains. (**B**) 15 DIV hippocampal neurons were stained with anti-PSD95 (magenta) and anti-synapsin-1 (green) antibodies. Immunolabeled objects were considered as synapses when both stainings were juxtaposed. Scale bar: 10 μm. Graph shows no difference in the number of synapses per μm$^2$ of Alix wt and ko neurons. (**C**) Brain sections from 8-week-old Alix wt or ko were stained by the Golgi–Cox impregnation technique. Stained dendritic segments were visualized by bright-field microscopy. Scale bar: 10 μm. Numbers of spines per μm of dendrites were counted. Average +/− SEM, $N$,

statistical analysis: (**A**) 0.40 +/− 0.02; 0.45 +/− 0.03 for Alix wt and Alix ko, respectively. $N$ = 600 synapses from 3 animals per genotype. Alix wt vs. Alix ko, $p$ = 0.1395, unpaired $t$ test. (**B**) 0.13 +/− 0.01; 0.12 +/− 0.01 for Alix wt and ko, respectively. $N$ = 7 independent experiments. Alix wt vs. Alix ko, $p$ = 0.5501, unpaired $t$ test. (**C**) 0.57 +/− 0.23; 0.51 +/− 0.01 for Alix wt and Alix ko, respectively. $N$ = 3 animals per genotype. Alix wt vs. Alix ko, $p$ = 0.1414, unpaired $t$ test. The data underlying all the graphs shown in the figure can be found in S1 Data. Alix, ALG-2-interacting protein X; DIV, day *in vitro*; ko, knockout; wt, wild type. (PDF)

**S6 Fig. Morphological and functional anomalies of Alix ko synapses.** (**A–D**) Representative electron micrographs of CA1 from Alix wt and ko mice. Scale bar: 200 nm. (**B, C, E, F**) Graphs represent the numbers of SVs per $\mu m^2$ (B), presynaptic bouton surface area (C), PSD length (E), ratio between the diameter of the spine head and neck (F) from sections throught the synapses. (**G**) LTP of fEPSP slope evoked by high frequency stimulation of Schaffer collaterals delivered at time 0. Inserts show representative EPSPs traces. Median (min to max), $N$, statistical analysis (B) 165.4 (69 to 278); 127.3 (31 to 336) for Alix wt and Alix ko, respectively. $N$ = 136 synapses from 3 animals. Alix wt vs. Alix ko, $p$ = 0.0009, Mann–Whitney test. (C) 0.13 (0.027 to 0.376); 0.16 (0.053 to 0.47) for Alix wt and Alix ko respectively. $N$ = 136 synapses from 3 animals. Alix wt vs. Alix ko, $p$ = 0.0036, Mann–Whitney test. (D) 0.76 (0.25 to 1.48); 0.54 (0.13 to 1.11) for Alix wt and Alix ko, respectively. $N$ = 136 synapses from 3 animals. Alix wt vs. Alix ko, $p$ = 0.0001, Mann–Whitney test. (E) 2.58 (0.85 to 5.60); 1.68 (0.58 to 4.78) for Alix wt and Alix ko, respectively. $N$ = 136 synapses from 3 animals. Alix wt vs. Alix ko, $p$ = 0.0001, Mann–Whitney test. (G) Average (10 last points): 182.9 +/− 6.68% and 130.7 +/− 2.08% for Alix wt and Alix, respectively. $N$ = 5 slices from 3 animals per genotype, $p$ = 0.0001, 2-way ANOVA. The data underlying all the graphs shown in the figure can be found in S1 Data. Alix, ALG-2-interacting protein X; EPSC, excitatory postsynaptic current; fEPSP, field excitatory postsynaptic potential; LTP, long-term potentiation; ko, knockout. (PDF)

**S7 Fig.** (**A**) Coronal brain section from control or Alix cko stained for DAPI (blue), GFAP (green), and Iba1 (red). Twenty-four-h post kainate injection. Scale bar = 500 μm. (**B**, **C**) Magnification of highlighted cortical (B) and hippocampal (C) areas show an increased ipsilateral (ipsi) microglial activation (Iba1) compared to contralateral (contra), which was more pronounced in Alix ko mice. Astroglial reactivity (GFAP) was moderately increased adjacent to the injection site in both groups. Scale bars = 100 μm. (**D**) Depiction of fluorescence intensity measurement areas in cortex, hippocampus, and hypothalamus. (**E**) In Alix ko mice, the contralateral Iba1 immunoreactivity was about 30% reduced in comparison to ipsilateral, in contrast to control mice. (**F**) GFAP immunoreactivities were not significantly different between hemispheres and experimental groups, as expected from the early analysis time point of only 24 h after kainate injection. Cortical GFAP immunoreactivity (A) was similar to hypothalamus (normalized fluorescence intensity ≈ 1). The hippocampal GFAP expression (B) showed a higher variability, however, approximately twice as high (normalized fluorescence intensity≈ 2) compared to cortex and hypothalamus. These data reflect the regional heterogeneity in astroglial GFAP expression, acting as internal confirmation of the analysis method. Circles and triangles represent individual quantifications from cortex and hippocampus, respectively. Large data points correspond to the average of 3 slices from the same animal (small data points in the background, color coded per animal). Fluorescence intensity values were normalized to the hypothalamic area (HT) of the respective hemisphere. Mean +/− IQR, N, statistical analysis: (E) Control: 1.1 (0.2) vs. 1.6 (0.4), contra- vs. ipsilateral medians (IQR); **$p$ = 0.009; Alix cko: 1.3 (0.5) vs. 1.5 (1.1), contra- vs. ipsilateral medians (IQR); $p$ = 0645. $N$ (control) = 4

animals, $N$ (Alix ko) = 3 animals. (F) control: 1.4 (1.1) vs. 1.5 (0.9), contra- vs. ipsilateral medians (IQR); $p$ = 0.6; Alix ko: 1.3 (1.1) vs. 1.7 (0.9), contra- vs. ipsilateral medians (IQR); $p$ = 0.6. $N$ (control) = 4 animals, $N$ (Alix ko) = 3 animals. The data underlying all the graphs shown in the figure can be found in S1 Data. Alix, ALG-2-interacting protein X; cko, conditional ko; IQR, interquartile range; ko, knockout.
(PDF)

**S1 Movie. Alix is recruited at presynaptic boutons upon stimulation.** Time laps recordings of 15–17 DIV hippocampal neurons expressing mcherry-Alix (left) and Syp-pH (right). While Alix is homogeneously distributed in non-stimulated neurons (0:00–2:00), Bic/4AP stimulation induced a relocalization of the protein to discrete puncta colocalizing with Syp-pH (2:20–7:00). Scale bar: 5 μm.
(MOV)

**S2 Movie. Endophilin accumulates at presynaptic boutons upon stimulation.** Time laps recordings of 15–17 DIV hippocampal neurons expressing endophilin-mcherry (left) and Syp-pH (right). Endophilin, which is present at presynaptic boutons under normal culture conditions (0:00–2:00), concentrates further at Syp-pH positive sites during Bic/4AP stimulation (2:20–7:00). Scale bar: 5 μm.
(MOV)

**S3 Movie. 3D reconstruction from FIB-SEM imaging of a cerebellar wt neuron synapse undergoing KCl-induced depolarization.**
(MOV)

**S1 Raw images. Full western blot from Fig 1A.** Uncropped, non-modified western blot films used in Fig 1A. Note that different exposures were used depending on the immunodetected proteins. **Full western blot from Fig 1B**. Uncropped, non-modified western blot films used in Fig 1B. Note that different exposures were used depending on the immunodetected proteins.
(PPTX)

**S1 Data. Excel sheets gathering the full set of raw data used in the different graphs presented in the manuscript.** Each excel sheet corresponds to 1 single panel and contains conditions, number of repeat, raw numbers, average, standard deviation, and statistical analyses.
(XLSX)

## Acknowledgments

We thank J. Brocard for helping with data analysis, Y. Saoudi for helping with fluorescence microscopy, N. Liaudet for his help with the segmentation and electron microscopy, and F. Saudou for making the MEA experiments possible. We also thank K. Sadoul for critically reading the manuscript. The authors are grateful to Paul Guichard and Virginie Hamel, without whom this publication would not have been possible.

## Author Contributions

**Conceptualization:** Marine H. Laporte, Rémy Sadoul.

**Data curation:** Marine H. Laporte, Kwang Il Chi, Laura C. Caudal, Na Zhao, Yvonne Schwarz, Marta Rolland, José Martinez-Hernandez, Magalie Martineau, Vincent Mercier, Florent Lemaître, Eve Moutaux, Maxime Cazorla, Fiona J. Hemming.

**Formal analysis:** Marine H. Laporte, Kwang Il Chi, Laura C. Caudal, Na Zhao, Yvonne Schwarz, Magalie Martineau, Eve Moutaux, Maxime Cazorla.

**Funding acquisition:** Rémy Sadoul.

**Methodology:** Christine Chatellard, Béatrice Blot.

**Software:** Eric Denarier.

**Supervision:** David Perrais, Fabien Lanté, Dieter Bruns, Sandrine Fraboulet, Frank Kirchhoff, Rémy Sadoul.

**Visualization:** Marine H. Laporte.

**Writing – original draft:** Marine H. Laporte, Fiona J. Hemming, Frank Kirchhoff, Rémy Sadoul.

**Writing – review & editing:** Marine H. Laporte, Frank Kirchhoff, Rémy Sadoul.

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
