## [Editor Report · Decision Letter 0]

22 Oct 2021

Dear Dr Laporte, 

Thank you for submitting your manuscript entitled "ALG-2 interacting protein-X (Alix) is required for activity-dependent bulk endocytosis at brain synapses" for consideration as a Research Article by PLOS Biology.

Your manuscript, reviews from Review Commons, and revision plan have now been evaluated by the PLOS Biology editorial staff as well as by an Academic Editor with relevant expertise and I am writing to let you know that we would like to consider a revised version of your manuscript that addresses the reviewer comments from Review Commons.

Before we can invite you to submit a revised manuscript, we need you to complete your submission by providing the metadata that is required for full assessment. To this end, please login to Editorial Manager where you will find the paper in the 'Submissions Needing Revisions' folder on your homepage. Please click 'Revise Submission' from the Action Links and complete all additional questions in the submission questionnaire.

Once you have completed your submission, we will send you a formal "major revision" decision, which will include additional comments from the Academic Editor and a 3 month deadline for the revision.

Please re-submit your manuscript within two working days, i.e. by Oct 26 2021 11:59PM.

Kind regards,

Lucas

Lucas Smith

Associate Editor

PLOS Biology

lsmith@plos.org

---

## [Editor Report · Decision Letter 1]

28 Oct 2021

Dear Dr Laporte,

Thank you for submitting your manuscript "ALG-2 interacting protein-X (Alix) is required for activity-dependent bulk endocytosis at brain synapses" for consideration as a Research Article at PLOS Biology. As mentioned in our last email, your manuscript, the reviews from Review Commons, and your revision plan have been evaluated by the PLOS Biology editors and an Academic Editor with relevant expertise.

In light of the reviews, which I have appended below, we will not be able to accept the current version of the manuscript. However, we would welcome re-submission of a much-revised version that thoroughly addresses the reviewers' comments. **IMPORTANT: Having discussed your revision plan with the Academic Editor, we think that it would be essential to experimentally confirm that differences in excitability between WT and KO neurons is not the reason for the ADBE phenotype reported here, as suggested by Reviewer 1 and emphasized by Reviewer 3. While we appreciate that this experiment will be difficult to perform, given the recent changes in your lab, we think these experiments would be necessary for further consideration.

We expect to receive your revised manuscript within 3 months. 

**IMPORTANT - SUBMITTING YOUR REVISION**

*Re-submission Checklist*

*Published Peer Review*

*PLOS Data Policy*

*Blot and Gel Data Policy*

Sincerely,

Lucas Smith

Associate Editor

PLOS Biology

lsmith@plos.org

REVIEWS:

---

## [Decision Letter · Decision Letter 2]

1 Apr 2022

Dear Dr Laporte,

Thank you for submitting your revised Research Article entitled "ALG-2 interacting protein-X (Alix) is required for activity-dependent bulk endocytosis at brain synapses" for publication in PLOS Biology. I have now obtained advice from the original reviewers and have discussed their comments with the Academic Editor. 

Based on the reviews, we will probably accept this manuscript for publication, provided you satisfactorily address address the following data and other policy-related requests (see below).

In addition, we would like you to consider a suggestion to improve the title:

"ALIX is required for activity-dependent bulk endocytosis at brain synapses"

We expect to receive your revised manuscript within two weeks. 

*Published Peer Review History*

*Press*

Sincerely,

Ines

--

Ines Alvarez-Garcia, PhD,

Senior Editor,

ialvarez-garcia@plos.org,

PLOS Biology

DATA POLICY:

Many thanks for complying with our data policy. Please also ensure that figure legends in your manuscript include information on WHERE THE UNDERLYING DATA CAN BE FOUND.

FINANCIAL DISCLOSURE

Please include grant numbers and the URLs of any funder's website.

We require the original, uncropped and minimally adjusted images supporting all blot and gel results reported in an article's figures or Supporting Information files. We will require these files before a manuscript can be accepted so please prepare and upload them now. Please carefully read our guidelines for how to prepare and upload this data: https://journals.plos.org/plosbiology/s/figures#loc-blot-and-gel-reporting-requirements 

Reviewers' comments

Rev. 1:

The authors have done a great job addressing my comments and I am happy to recommend acceptance. There is just one thing - Figures 3K and S3C look identical to me, can these be checked please?

Rev. 2: Charles A Hoeffer – note that this reviewer has signed his review

This reviewer is satisfied with the responses to the reviewer critiques (mine and others). I appreciate the time and extra effort to include the additional hyper excitability experiments and the reworking of the statistics to conform to more rigorous norms. The publication is acceptable for publication and is of greater interest than the previous submission.

Rev. 3: Silvio O. Rizzoli – note that this reviewer has signed his review

The authors have replied to all of my comments. I suggest that the manuscript be published in its current form.

---

## [Editor Report · Decision Letter 3]

4 May 2022

Dear Dr Laporte,

On behalf of my colleagues and the Academic Editor, Thomas Südhof, I very happy to say that we can in principle accept your Research Article entitled "Alix is required for activity-dependent bulk endocytosis at brain synapses" for publication in PLOS Biology, provided you address any remaining formatting and reporting issues. These will be detailed in an email that will follow this letter and that you will usually receive within 2-3 business days, during which time no action is required from you. Please note that we will not be able to formally accept your manuscript and schedule it for publication until you have completed any requested changes.

PRESS

Many congratulations and thank you again for choosing PLOS Biology for publication and supporting Open Access publishing. We look forward to publishing your study. 

Sincerely, 

Ines

--

Ines Alvarez-Garcia, PhD 

Senior Editor 

PLOS Biology
